# Haplosaurus computes protein haplotypes for use in precision drug design

William Spooner[1,2], William McLaren[3], Timothy Slidel[4], Donna K. Finch[4], Robin Butler[4], Jamie Campbell[4], Laura Eghobamien[4], David Rider[4], Christine Mione Kiefer [5], Matthew J. Robinson[4], Colin Hardman[4], Fiona Cunningham [3], Tristan Vaughan[4], Paul Flicek [3] & Catherine Chaillan Huntington [4]

Selecting the most appropriate protein sequences is critical for precision drug design. Here we describe Haplosaurus, a bioinformatic tool for computation of protein haplotypes. Haplosaurus computes protein haplotypes from pre-existing chromosomally-phased genomic variation data. Integration into the Ensembl resource provides rapid and detailed protein haplotypes retrieval. Using Haplosaurus, we build a database of unique protein haplotypes from the 1000 Genomes dataset reflecting real-world protein sequence variability and their prevalence. For one in seven genes, their most common protein haplotype differs from the reference sequence and a similar number differs on their most common haplotype between human populations. Three case studies show how knowledge of the range of commonly encountered protein forms predicted in populations leads to insights into therapeutic efficacy. Haplosaurus and its associated database is expected to find broad applications in many disciplines using protein sequences and particularly impactful for therapeutics design.

[1] Eagle Genomics Ltd., Biodata Innovation Centre, Wellcome Genome Campus, Hinxton, Cambridge CB10 3DR, UK. [2] Genomics England, QMUL Dawson Hall, London EC1M 6BQ, UK. [3] European Molecular Biology Laboratory, European Bioinformatics Institute, Wellcome Genome Campus, Hinxton, Cambridge CB10 1SD, UK. [4] MedImmune Ltd., Granta Park, Cambridge CB21 4QR, UK. [5] MedImmune, Gaithersburg, MD 20878, USA. These authors contributed equally: William Spooner, William McLaren, Timothy Slidel. Correspondence and requests for materials should be addressed to D.R. (email: d.rider@silence-therapeutics.com) or to C.C.H. (email: HuntingtonC@medimmune.com)

Proteoforms are the different molecular forms in which the protein product of a single gene can be found[1]. Proteoforms modulate a wide variety of biological processes and contribute to many phenotypes and diseases[2]. There are two main classes of proteoforms; protein isoforms, which include alternatively spliced RNA transcripts and post-translational modifications, and protein haplotypes, where protein changes are due to genomic variation. Whilst there are many genome-wide tools and databases for protein isoforms[3,4] and genomic variation[5,6], there is, with rare exceptions[7], an almost complete absence of resources for protein haplotypes. We are addressing this gap in the work presented here. We focus this paper on population-level, genome-wide distributions of common protein haplotypes that can potentially impact drug binding, rather than the specific (and often rare) haplotypes that cause diseases or act as marker proteins.

An individual's genome is diploid comprising both the maternal and paternal allelic sequences, i.e., each gene has two haplotypes. We define a protein haplotype as the translation of a spliced RNA transcript derived from a gene haplotype. In diploid genomes, function is mediated by two forms of the gene/protein, i.e., by pairs of haplotypes (diplotypes)[7]. The diploid nature of the human genome, despite being fundamental to protein function, is often ignored in genomic studies[8]; we demonstrate how this omission could cause problems in drug development. The groundwork for protein haplotype and diplotype architecture of human genomes was laid by Hoehe et al.[7] who described a first systematic, quantitative, population-based analysis of protein haplotypes. Our work described here extends this earlier work in scale, ease of access, availability of tools for analysis and in concrete applicability to a medically relevant subject.

Differences in haplotypes and diplotypes of the protein target can result in variable therapeutic responses to drug treatment[9,10]. To modify the activity of a disease-associated protein target, drug discovery programmes aim for a drug molecule optimised to be cross-reactive to the widest spectrum of target proteoforms in the patient population. This optimisation improves therapeutic response[11,12], achieving one of the key aims of precision medicine[13]. The diversity of protein target haplotypes is of particular importance for biologic drugs, such as monoclonal antibodies (mAbs), which display exquisite specificity for their target protein sequence. MAbs now account for almost 40% of drugs in clinical development[14], and their number is increasing[15].

As it is not feasible to synthesise and experimentally validate more than a handful of target proteins for use in optimisation of a new drug, a scientist will prioritise the protein sequence to be used in their drug development strategy according to two main criteria: structure–function relation of the variants and their prevalence. First, all potential protein-altering genetic variants in the target are assessed using tools such as Ensembl's Variant Effect Predictor[16]. Those with potential impact on drug binding, using knowledge of the target domains involved in pharmacologically relevant interactions, are prioritised for the selection and screening strategy. Secondly, the frequency at which each unique protein haplotype occurs within a population is an important measure of potential clinical impact. We use frequency of occurrence, FoO (following the terminology of Hoehe et al.[7]), calculated as the count of a unique haplotype of a gene in a population divided by total haplotype count (twice the number of individuals in the population). Only haplotypes with FoO equal to or above some threshold, typically 1%, are considered to be clinically relevant to drug development. Frequency-based protein haplotype selection is commonly performed as shown in Fig. 1. Although the target diplotype rather than haplotype is the ultimate determinant of activity of a drug in an individual, only the haplotype needs be considered, as a drug effective for 99% of protein haplotypes in a population is statistically guaranteed for 98% of diplotypes.

Historically, frequency-based protein haplotype selection in drug development has relied on in-house DNA sequencing of the target gene in tens to hundreds of individuals. Recently, large population sequencing projects such as the 1000 Genomes project have revolutionised the analysis of human genetic variation, potentially alleviating the need for costly bespoke cohort sequencing in therapeutic projects. Estimation of haplotypes from genetic variation data uses statistical imputation strategies based on genotype population frequencies, known as phasing (Fig. 1a–c). Phased genotypes for the 1000 Genomes project are readily available. A criticism of imputation-based phasing methods such as those used for 1000 Genomes[18] is that they are ineffective for rare and de novo variants[19]. However, in drug discovery it is the common haplotypes that are most significant, and these are more likely to be based on reliable variant frequency information needed for imputation. Indeed, it has been shown, using data from the 1000 Genomes Project, that there is high concordance between protein haplotype sequences generated from statistically phased (short read) data and from molecularly phased (long-read) data, supporting the use of statistically phased data in addressing protein haplotype architecture within populations[7]. The accuracy of population-based phasing is likely to increase as more detailed imputation panels, such as from the Haplotype Reference Consortium[20], become available.

Despite the availability of phased genotype data and the importance to drug discovery of understanding protein haplotypes, we have found no methods that can compute protein haplotype data from phased genotypes or resources that make precomputed protein haplotypes available. Instead, existing resources such as dbSNP[21], Ensembl[22], UniProt[23] and RefSeq[24] are limited to the display of individual variants within each gene/isoform/protein, with population frequency shown as the minor allele frequency (MAF) for the individual variants. To inform drug discovery, we need information on how alleles are phased into protein haplotypes alongside the population frequencies of the resulting proteoforms. Haplosaurus addresses these needs by combining (Fig. 1d–h) reference genome sequence, gene models and phased variation data (e.g., from 1000 Genomes project) to present a comprehensive overview of predicted protein haplotypes and their frequencies across populations.

In this paper, we seek to address the lack of accessible tools and resources for protein haplotypes in three ways: (i) developing "Haplosaurus", a bioinformatics tool for computing protein haplotype sequences from pre-existing genotype data; (ii) using Haplosaurus to build a database of protein haplotypes from the 1000 Genomes dataset and (iii) contributing new Haplosaurus-based views to the Ensembl website (www.ensembl.org) to provide convenient and rapid access to protein haplotypes on a per-gene/isoform basis. From our database, we analyse the landscape of protein haplotype variability; between populations, between druggable gene classes, and for clinical mAb targets, between clinical trial phases or market status. Through three case studies, we demonstrate the potential impact of Haplosaurus on drug discovery through the identification of relevant protein haplotypes for greater patient coverage. The benefits of Haplosaurus for protein scientists are summarised in Fig. 1.

## Results

**Haplosaurus software for computing protein haplotypes.** Haplosaurus is an open-source software built on the existing Ensembl codebase (as described in Methods). The availability of a robust database and Application Programming Interface (API) for manipulating gene models, genomic variants and variant call

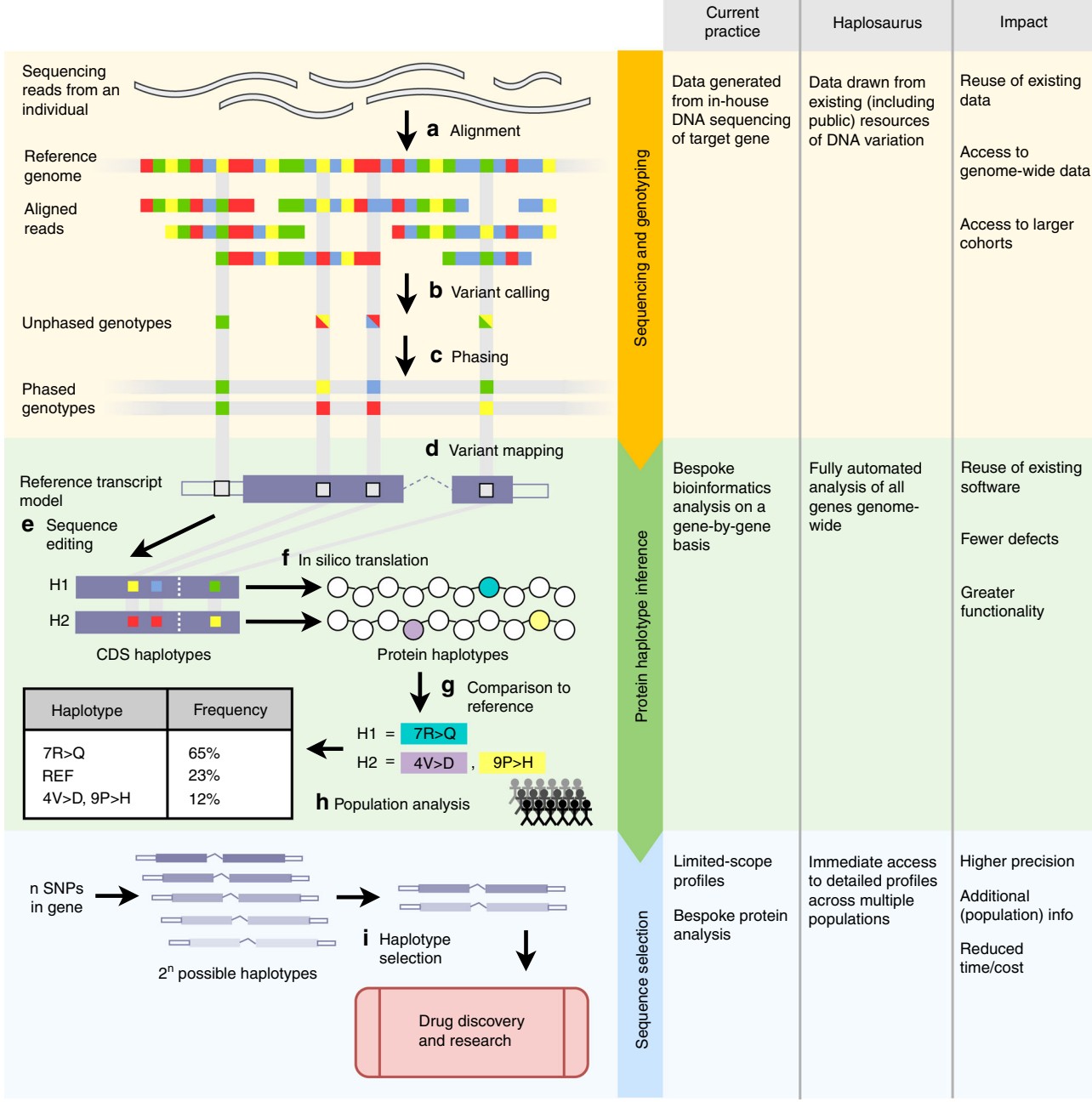

**Fig. 1** Overview of inference and selection of protein haplotypes. **a** Raw sequencing read data obtained from a single individual are aligned to the reference genome of the target region. **b** Variants are 'called' by detecting loci where the reads differ from the genome, giving a set of genotypes across the target. **c** Heterozygous genotypes are assigned into separate parental haplotypes using established statistical phasing methods[17]. **d** Variant loci are mapped relative to reference transcript models. **e** Phased variant alleles are used to edit the reference coding sequences (CDSs), generating two CDS haplotypes for the individual, one for each phase. **f** CDS haplotype sequences are translated in silico to produce corresponding protein haplotypes. **g** Haplotype sequences are aligned and compared with the reference translated protein sequence, and the differences are summarised using a simple nomenclature. **h** Haplotypes generated are collated across many sequenced individuals to generate population frequency data. **i** The prevalence of each haplotype sequence is used to inform the selection of protein sequence(s) to be used as a target in drug discovery. Benefits of Haplosaurus versus current state of art for inference and selection of protein haplotypes is described on the right panel

format (VCF) files along with an active collaborative development community made Ensembl the ideal platform for Haplosaurus development. Haplosaurus can be used with any of the 100 s of species for which there is an Ensembl database. The software has been validated with simulated sequence data generated using independent software (as described in Methods).

For a given gene identifier and VCF file, Haplosaurus retrieves the corresponding gene model from a linked Ensembl database.

Next, the phased variants that overlap the gene's location are retrieved from the VCF file and used to generate two lists of DNA sequence alleles for each sample, one for each DNA haplotype, according to their phase. The DNA haplotype sequences are reconstructed by substituting the alleles for each haplotype into the reference sequence according to their genomic location. The DNA haplotypes are virtually 'transcribed' to coding sequence (CDS) haplotypes, and these are then virtually 'translated' into

protein haplotypes. Once the protein haplotypes for many samples have been generated, the software can calculate the FoO of each unique protein haplotype sequence in a population.

We use a notation for protein haplotypes similar to that of the Human Genome Variation Society (HGVS), where differences to the reference sequence are combined in position order with the transcript or protein identifier. For example, ACTN3-001:211R>Q, 577R>*, 578del(325) indicates, for the ACTN3-001 transcript, an R to Q substitution at position 211, followed by the introduction of a stop codon at position 577 that results in the truncation of the following 325 residues of the protein.

**A protein haplotype collection from the 1000 Genomes dataset**. To enable genome-wide analysis of protein haplotype diversity, we used Haplosaurus to build a database of unique protein haplotypes from phased haplotypes imported directly from the 1000 Genomes Project phase 3 VCF file (as described in Methods). Our database comprises protein haplotypes for each of the 20,166 human protein-coding genes for 2504 individuals from five superpopulations: African, Admixed American, South Asian, European and East Asian. The data are available on a per-gene basis via the Ensembl website (as described in Methods, see Supplementary Fig. 1 for an example). Although our database contains protein haplotypes for each alternatively spliced transcript (isoform) annotated by Ensembl, we have, for simplicity, restricted our analysis here to a single canonical isoform for each gene (as described in Methods).

In our database, the number of unique protein haplotypes per gene ranges from 1 to 4554, with an average of 35.7 across all 20,166 genes. Our calculation includes all haplotypes, including the reference sequence (where it occurs), and all genes, both variable and invariable. A table of unique haplotype counts for each gene is included in Supplementary Data 1. Haplotype counts are likely to be reliable for most genes, but overestimated for genes in regions where phasing is difficult (e.g., low-linkage disequilibrium) or variant calling is error prone (e.g., repetitive regions). Notwithstanding, a gene's reference protein is often not its most common protein form: for one in seven genes (3512 genes), the most common protein haplotype does not correspond to the Ensembl protein sequence inferred from the reference genome sequence. For 219 of these genes (~1% overall), this reference protein is not seen at all among the 1000 Genomes haplotypes. We have also compared our protein haplotype collection with UniProt and RefSeq (as described in Methods), which often provide the reference protein sequences used in drug development. A total of 16,973 genes have a protein haplotype that maps exactly to a UniProt entry and 18,520 to a RefSeq. However, for one in six (2731 genes) where at least one haplotype maps to UniProt, the most common haplotype does not map, with the number being one in seven (2749 genes) for RefSeq.

The entirety of unique haplotypes/diplotypes increases in relation to increasing number of genomes[7]. In our database of 2504 genomes across 20,166 genes (a total of 100,991,328 haplotype observations and 50,495,664 diplotype observations), we observe a total of 718,964 unique protein haplotypes and 1,068,742 unique protein diplotypes. To account for sample size, we limit many of our analyses to common protein haplotypes that we define as those that equal or exceed a threshold frequency of occurrence (FoO) of 1%. The total number of protein haplotypes, FoO ≥ 1%, in our database is 55,108, an average across all genes of 2.7 haplotypes per gene. For diplotypes we observe 73,680 with FoO ≥ 1% (3.6 per gene). Supplementary Data 1 includes counts of common protein haplotypes for each gene.

Following Hoehe et al.[7], we classify genes into distinct categories based on their number of common haplotypes. This provides an indication of protein haplotype complexity, which is a consideration in drug design. For gene targets with only a single common protein haplotype (FoO ≥ 1%), protein variability is generally not a consideration; just under two-fifths of all genes (7708 genes) fall into this category. A similar number (7671 genes) have two or three common haplotypes, and a quarter (4785 genes) have four or more common haplotypes from which a smaller set would be prioritised and selected for drug optimisation.

**Protein-altering DNA variants and their protein haplotypes**. Each protein haplotype results from a distinct combination of protein-altering DNA variants (polymorphisms) in a given gene. The relationship between DNA variants and resultant protein haplotypes is confounded by linkage disequilibrium, and there is no simple transformation between the two i.e., protein haplotype frequency is not simply the product of the frequencies of its constituent variants. To illustrate the complexity of the relationship, we compare (Fig. 2) the number of common (MAF ≥ 1%) protein-altering variants per gene against the corresponding number of common (FoO ≥ 1%) protein haplotypes (Supplementary Software 1[25]). Whilst the number of haplotypes generally increases alongside the number of protein-altering variants, some genes have relatively fewer haplotypes-per-variant and vice versa. Proteins identified with a high number of variations per haplotype (Fig. 2a) include the three major types of human MHC class 1 cell surface receptors, HLA-A, B and C, where sequence diversity plays a key functional role[26]; and four mucin proteins, including MUC-4, with the highest number of protein haplotypes are found in this analysis. Mucins are recent paralogues and contain stretches of variably repeated coding sequences, which makes sequence alignment and variant calling error prone[27,28]; it is probable that the extreme apparent divergences between haplotypes in these genes are artefacts of misalignment rather than truly high frequencies of missense (substitutional) variation.

**Protein haplotypes and population ancestry**. We compared the protein haplotype variability of the five 1000 Genomes super populations across all genes (Fig. 3a and Supplementary Data 2). The average number of common protein haplotypes per gene is greatest in Africans (3.0 per gene) and lowest in East Asians (2.0 per gene). This is a statistically significant difference with a p-value less than 2.2e10-16 (Mann–Whitney test). Similarly, the number of genes with four or more common protein haplotypes is double in Africans (27% of genes) compared with East Asians (12% of genes). Africans have greater protein haplotype diversity and is consistent with previous observations of overall genetic diversity[29], and has implications for the development of drugs for the African population[30].

We find that population-level differences in protein haplotype frequency potentially result in different target proteins being selected for research or drug discovery purposes. For instance, the single most common protein haplotype (the protein most likely to be selected for drug optimisation), differs across the five superpopulations in 17% (3495) of genes. The set of common protein haplotypes (those most likely to be considered for drug screening) differs across the superpopulations in 74% (14,920) of genes. Finally, almost half (3193 genes) of the 7708 genes that have only a single common haplotype globally actually have two or more in at least one population. Based on the global analysis, these would not generally be considered for further haplotype investigation without this additional population information.

**Protein haplotypes and biopharmaceuticals**. Putative druggable genes are of particular importance in the context of drug

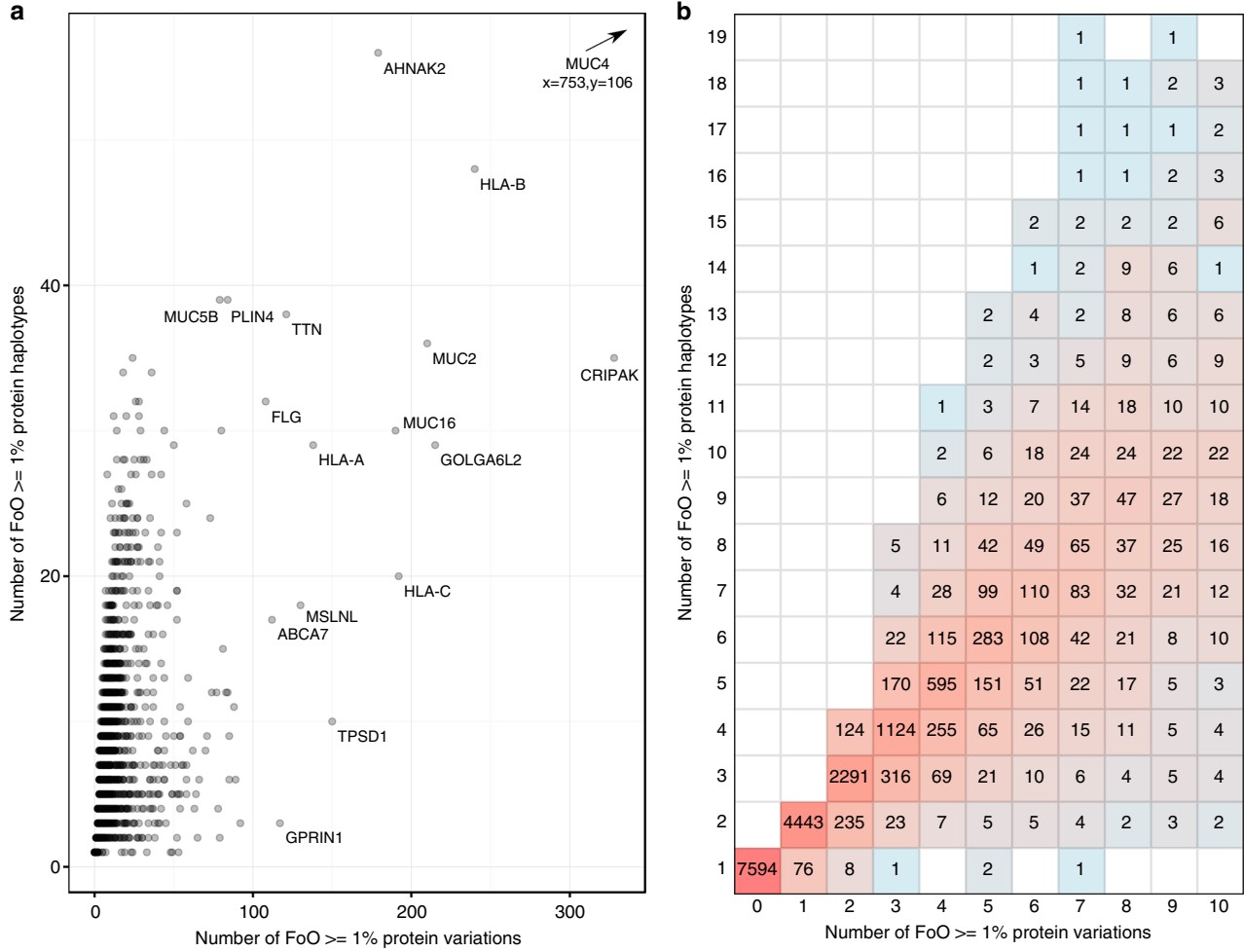

**Fig. 2** Variations in a protein and number of protein haplotypes relationship. Variations and haplotypes have frequency FoO >= 1%, see Methods: Variants vs. haplotypes—analysis for Fig. 2 for further information. **a** Scatter plot for all proteins. Gene symbols are shown for some example proteins with high numbers of variations and haplotypes such as *MUC4* which is positioned outside of the plot area in the top right corner. **b** 2D histogram showing counts of proteins with < 11 variations and < 20 haplotypes, the red-blue colour scale represents the natural log of the count

discovery. A total of 9163 druggable genes in 40 categories have been annotated by the Drug Gene Interaction Database (DGIdb v2.22)[31]. Genes inside the druggable genome show on average a greater number of common protein haplotypes than those outside (3.0 and 2.7 per gene, respectively). This difference is statistically significant, p-value less than 2.2e10-16 (Wilcoxon rank sum). Figure 3b (and Supplementary Data 2) shows the haplotype frequency distribution across five categories of druggable genes that are particularly amenable to biologic drugs. In each category other than transporter, they have more haplotypes than the overall average, with a particularly increased average of 3.4 haplotypes per gene for G Protein Coupled Receptor genes (GPCRs). Haplotype counts for genes in all 40 DGIdb categories can be found in Supplementary Data 3.

Biotherapeutic targets with a high number of common protein haplotypes represent a potential risk to clinical efficacy within the intended patient population. To assess the current Biologics in clinical phase, we analysed protein haplotypes of the 221 different therapeutic targets either approved or in registered clinical trials as of October 2016 (source: Informa Pharmaprojects). The same targets are pursued by multiple companies and evaluated in several indications amounting to a total of 606 biopharmaceuticals programmes. Of these 221 genes, 144 had two or more common protein haplotypes, with an average of 3.1 per gene

(Fig. 3c and Supplementary Data 2). There are, therefore, a sizeable number of drugs that are in development or being marketed, for which the population coverage (in terms of activity across all haplotypes) is potentially suboptimal. Although there is potential for suboptimal coverage, we find no statistically significant trends in target protein haplotype complexity with trial phase across this relatively small set of genes.

The direct impact of protein haplotype variability on drug development programmes or on the clinical efficacy of marketed drugs is rarely reported. The paucity of published reports does not necessarily mean that impact is uncommon. Here we present three example gene targets; one from the literature, *C5*, and two from our in-house drug discovery pipeline; *TLR4* and *FPR1*. These examples are contrasting in approaches used to manage haplotype variability; *C5* not at all, *TLR4* once a development problem arose and *FPR1* to avoid any problem from the start. In each case a priori data from Haplosaurus would have been valuable.

**Haplosaurus case study 1 using C5 and eculizumab.** Eculizumab is a mAb that is highly effective for the treatment of paroxysmal nocturnal haemoglobinuria (PNH). However, variation of the protein target (885R>H in complement protein *C5*) within

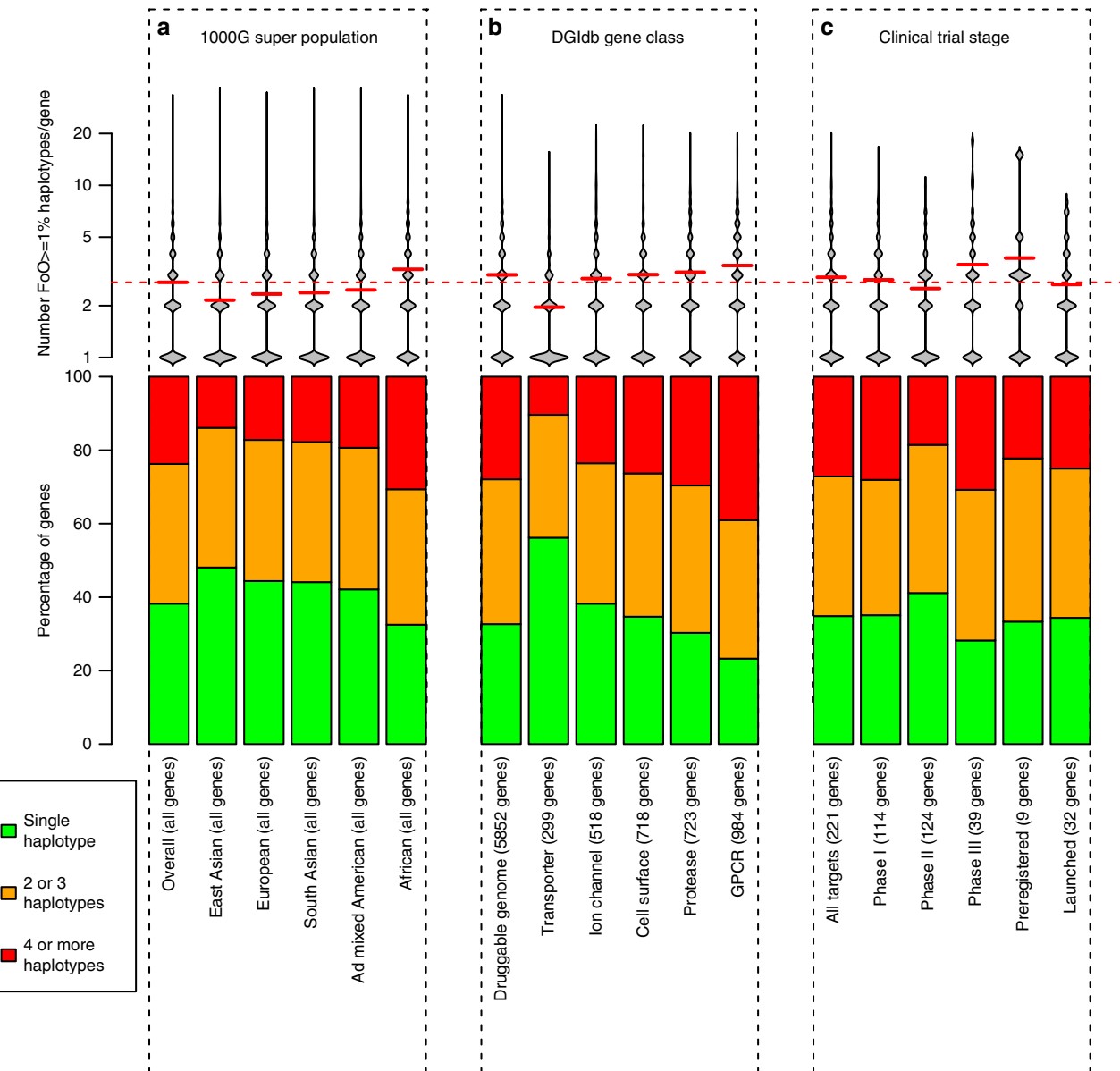

**Fig. 3** Distributions of protein haplotypes per gene in the 1000 Genomes. For the bean plots (top row) the distribution of number of protein haplotypes (FoO ≥ 1%) per gene is represented as a density shape (Sheather & Jones kernel density method) with the average value for each group indicated by a solid red line, and the overall average by a dashed red line. The stacked bar charts (second row) show the percentage of genes having a single significant protein haplotype, 2 or 3 significant haplotypes, or > 4 significant haplotypes (where the level of significance is FoO ≥ 1%). In **a**, the number of haplotypes per gene is computed overall, and for each 1000 Genomes ancestral super-population. In **b,** the overall 1000 Genomes population is used and genes are grouped by druggable gene category as defined by the Drug Gene Interaction Database (DGIdb) and selected for their relevance to mAb development. **c** shows distributions for genes targeted by therapeutic mAb programmes, grouped according to their clinical trial stage or market status. Genes within any group are unique, but a single gene may be counted in multiple groups

the epitope binding site has been reported to result in loss of efficacy in 3.5% of Japanese patients[32]. We compared Haplosaurus results with this observation. A total of 86 protein haplotypes for the *C5* gene are seen in our 1000 Genomes database. Of these, 12 are common (FoO ≥ 1%) in at least one super-population. Frequencies of individual haplotypes vary greatly between populations (Fig. 4). Of particular interest is C5:802V>I,885R>H, the haplotype that harbours the 885 R > H mutation responsible for the loss of eculizumab activity. With a frequency globally of just 0.2%, it would be considered insignificant for drug development. It is only found in the east asian superpopulation (FoO = 0.9%), and for east asians mainly within

the Japanese population where it occurs with a frequency of 3.4%. In all 1000 Genomes carriers or C5:802V>I,885R>H, the haplotype is heterozygous, suggesting that a single copy causes complete loss of efficacy. Thus, our resource correctly identifies even subtle cases of protein diversity that can affect the efficacy of marketed drugs. This case also demonstrates loss of efficacy affecting appreciable numbers of a specific patient population even when the global haplotype frequency is well below the 1% threshold.

**Haplosaurus case study 2 using TLR4 and MEDI-2843.** Toll-like Receptor 4 (*TLR4*) was the target of MEDI-2843, an

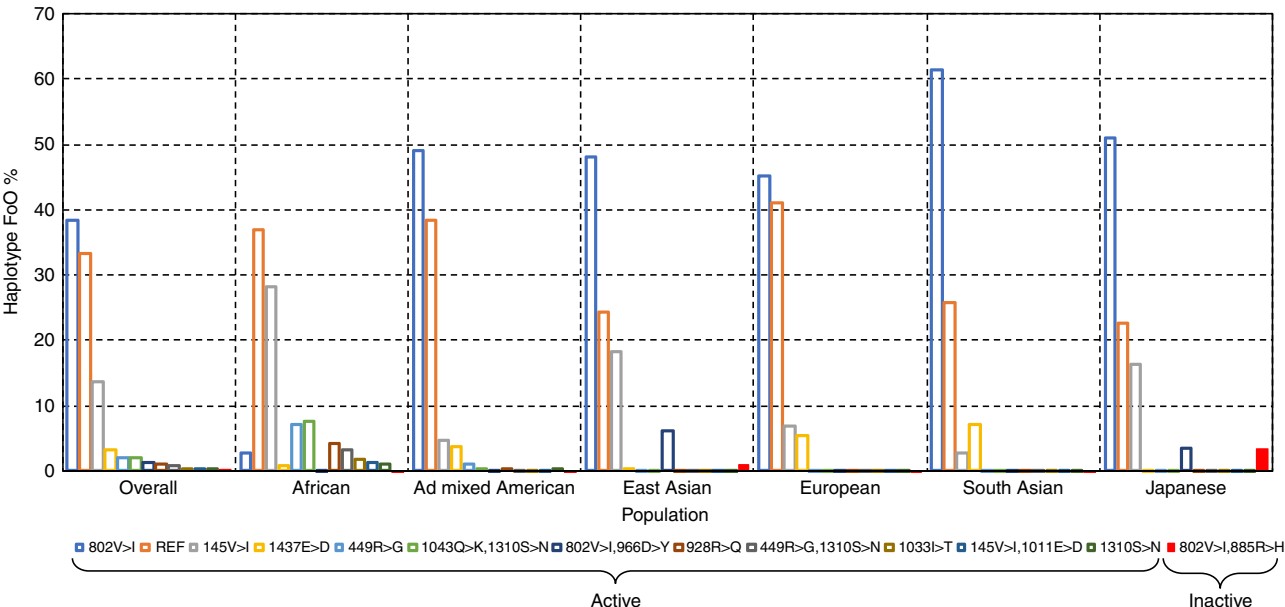

**Fig. 4** Frequency of *C5* protein haplotypes by population. Haplotypes are included in the plots if their frequency is ≥ 1% in one or more of the populations. The haplotype C5:802V>I,855R>H, which results in loss of activity of eculizumab is highlighted in red. Data for *C5* are taken from http://grch37.ensembl. org/Homo_sapiens/Transcript/Haplotypes?t=C5-001

antagonistic mAb aimed at reducing innate inflammatory response in patients with chronic inflammation. During drug development, the *TLR4* reference protein (UniProt O00206) was selected as the sole antigen for MEDI-2843 isolation. While effective in inhibiting *TLR4*-mediated responses in cells derived from 14 out of 17 anonymous human donors, MEDI-2843 demonstrated only a partial response in the remaining three (Supplementary Fig. 2a). In-house DNA sequencing determined that the three donors with incomplete response were heterozygous for haplotype TLR4:299D>G,399T>I. Binding of MEDI-2843 was abolished entirely in cells engineered to overexpress TLR4:299D>G (Supplementary Fig. 2b). This explained the findings from the donor cell experiments and clarified previous conflicting reports on the effect of the TLR4:299D>G on LPS binding and signalling[33,34].

A retrospective analysis of the protein haplotypes distribution for *TLR4* generated using Haplosaurus allows us to examine the MEDI-2843 case in more detail (Fig. 5). Three common protein haplotypes carry the 299D>G variant responsible for loss of MEDI-2843 activity, with a combined frequency of 6.2%. For each protein haplotype, it is possible to access in the table of population frequencies the detailed information for each individual if that haplotype is seen once (heterozyguous) or twice (homozygous). Analysing diplotype distributions we see, in the full 1000 Genomes set, 12.1% of individuals have at least one 299D>G haplotype copy (reduced activity of MEDI-2843) and 0.4% have two (complete loss of MEDI-2843 activity). The 299D>G haplotypes frequencies vary greatly between populations, being lowest in East Asians (under 1%) and highest in South Asians (over 13%). The most common 299D>G-carrying haplotype also differs, being TLR4:299D>G,399T>I in most populations, but TLR4:299D>G in Africans. Had such information on *TLR4* haplotype diversity been available at the outset, it is likely that the drug would have been designed against proteins carrying TLR4:299D>G, thus saving considerable time and effort. This example demonstrates that assessing not only the overall protein haplotype frequency but also their diplotypes, as enabled by Haplosaurus, can provide important insight.

**Haplosaurus case study 3 using FPR1 and mAb Fpro0155**. Fpro0155 is an antagonistic mAb to formyl peptide receptor 1 (*FPR1*), a class A GPCR that mediates inflammatory response driven by leucocytes, particularly neutrophils[35]. Fpro0155 was designed to reduce chronic pathogenic neutrophilic inflammation. At the time five non-synonymous single-nucleotide polymorphisms (SNPs) had been reported[36] within *FPR1*. To evaluate the drug's activity against all SNP combinations would have required engineering and testing of 32 different protein haplotypes, which would have been costly and time consuming. The alternative, to test the drug on common SNP combinations (i.e., common haplotypes), required determination of their frequencies. This was achieved using sequence-based genotyping of the *FPR1* locus in 65 unrelated UK donors (European Collection of Authenticated Cell Cultures). This resulted in 13 common protein haplotypes with FoO ≥ 1% (Fig. 6a), of which the most prevalent, 11I>T,192N>K,346E>A (FoO = 24%) was used to derive Fpro0155. The haplotype data were further used to select four proteins that encompassed all SNPs in extracellular, intracellular and transmembrane domains of the *FPR1* molecule (Fig. 6a). This allowed Fpro0155 activity to be tested against protein haplotypes representing over 98% of the *FPR1* extracellular diversity. Binding of Fpro0155 to all four was confirmed (Supplementary Fig. 3a), as was functional inhibition in a calcium signalling assay using engineered cell lines (Supplementary Fig. 3b). Here, using Haplosaurus streamlined and focused efforts on four real-world protein sequences relevant to the therapeutic development instead of having to consider the 32 putative different ones. The benefit of using Haplosaurus for haplotype analysis becomes visible already when the therapeutic target displays more than two potential sequence variations. This example highlights how protein haplotype analysis is powerful and provides additional information, compared with analysis of individual amino acid variants, for drug design.

As an alternative to in-house sequencing, comparable information on *FPR1* protein haplotype diversity can now be obtained via the Haplosaurus/Ensembl Transcript Haplotype View, http://grch37. ensembl.org/Homo_sapiens/Transcript/Haplotypes?t=FPR1-003.

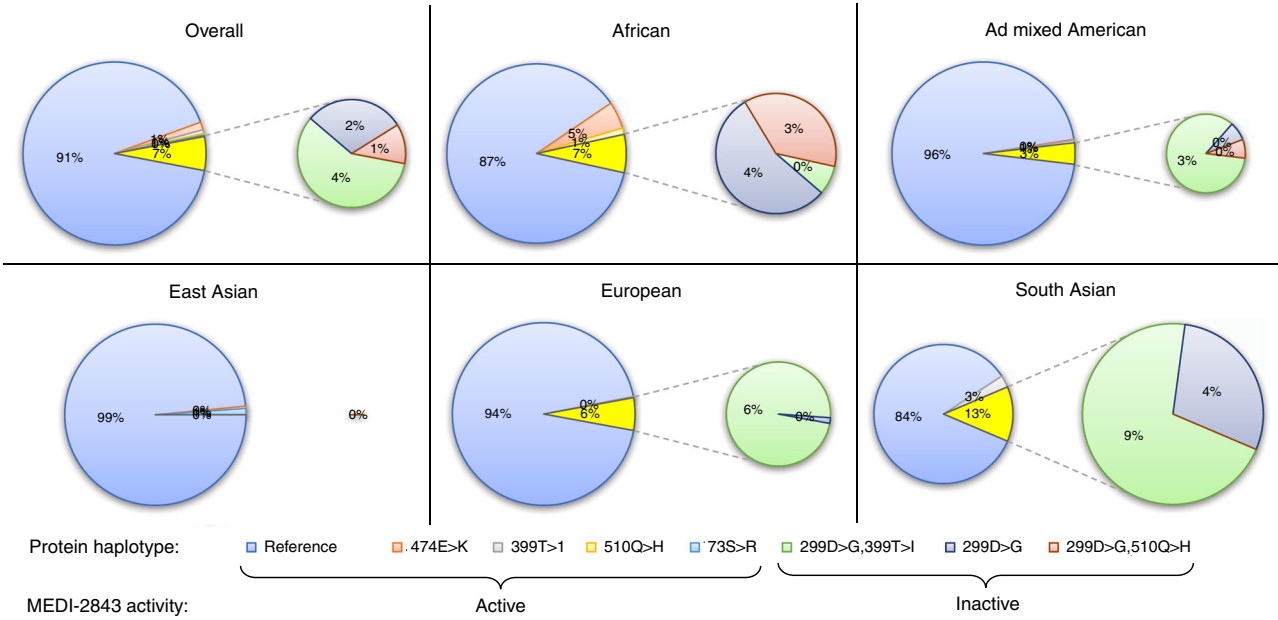

**Fig. 5** Frequency of *TLR4* protein haplotypes (FoO ≥ 1%) by population. The haplotypes TLR4:299D>G,399T>I, TLR4:299D>G and TLR4:299D>G,510Q>H that carry 299D>G and result in loss of activity of MEDI-2843 are expanded on the right. Data are taken from http://grch37.ensembl.org/Homo_sapiens/Transcript/Haplotypes?t=TLR4-001

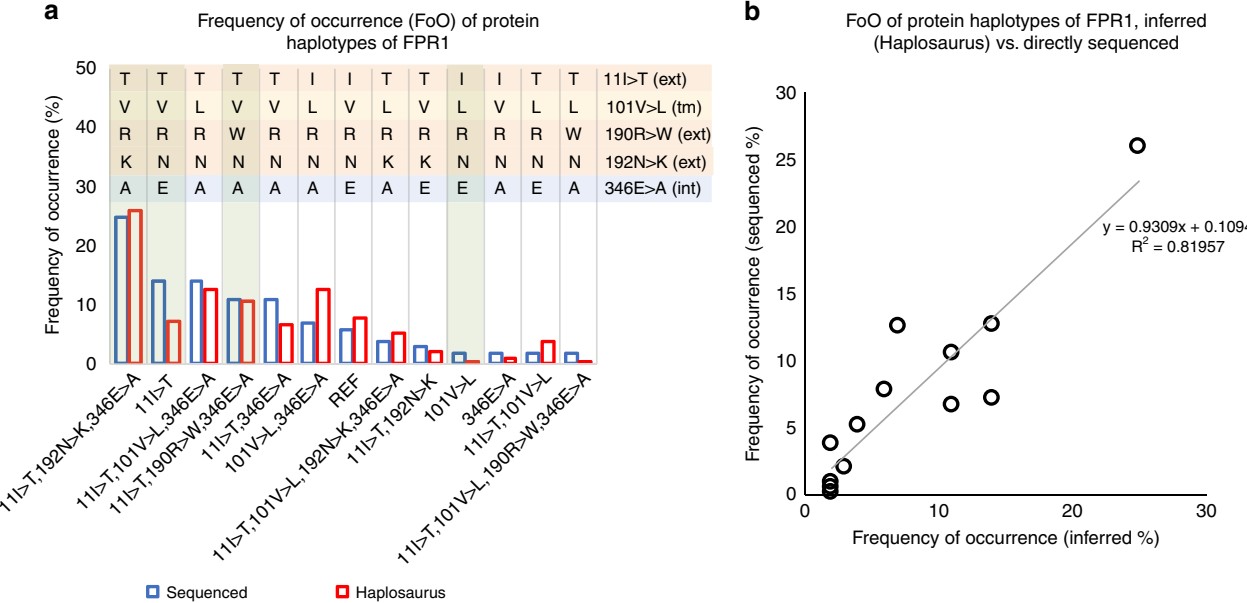

**Fig. 6** Frequencies of the thirteen FPR1 protein haplotypes. The frequency for the thirteen FPR1 protein haplotypes sequenced by MedImmune and the frequency in the 1000 Genomes European population inferred by allele frequency and by the Haplosaurus method are compared. **a** The four haplotypes for which binding and inhibitory activity of Fpro0155 was confirmed experimentally are highlighted in green. The five non-synonymous SNPs (following the numbering used for UniProt P21462) are highlighted at the upper right according to their topological localisation: extracellular (ext) in orange, transmembrane (tm) in yellow and intracellular (int) in blue. Letters represent amino acid abbreviations. **b** correlates the haplotype frequency determined by sequencing with allele frequency inferred by Haplosaurus

There is a good correlation ($r^2 = 0.82$) between *FPR1* haplotype frequency inferred by Haplosaurus vs. frequency from the earlier direct sequencing project (Fig. 6b). Furthermore, Haplosaurus values are likely more representative as they are derived from a larger population (503 individuals) versus 65 individuals for direct sequencing. In the case of *FPR1*, the benefits of Haplosaurus for research and for drug discovery become obvious; in reduction of

haplotype analysis time, in increased confidence of data from a larger sample and in relevance to multiple populations.

## Discussion
Selecting the most appropriate protein sequences for a given application is not trivial; for a considerable number of genes (~1

in 7) the genomic reference, or reference proteins represented in UniProt or RefSeq, are not the most common haplotype. Haplotype distributions also vary significantly between human populations and gene classes. Our work, available through the Ensembl Transcript Haplotype View or via download of free access data and software, provides convenient and rapid access to protein haplotype frequencies across multiple human populations. This allows scientists to profile a gene's naturally occurring protein forms, and to prioritise and select the most appropriate (set of) protein sequences for a given application. This is important in many areas of protein research, including in drug design to achieve optimal population coverage. Furthermore, information from protein haplotypes and their diplotype pairs enables the functional importance of phase to be assessed, information that is ignored in the analysis of individual protein-coding variants.

We have shown that some common classes of druggable proteins have increased protein haplotype diversity compared with human proteins overall, and that antibody drugs exist, both marketed and in clinical trials, whose target proteins have high haplotype diversity. Drug discovery for these began when Precision Medicine and patient responder sub-sets were early concepts and this raises questions regarding the impact of protein haplotype diversity on drug efficacy. The risk is exemplified by the loss of efficacy of eculizumab in a Japanese population carrying the C5:802V>I,885R>H haplotype of the C5 gene.

Precision medicine aims to increase the efficacy of drugs by considering individual genetic variability on drug response (pharmacogenomics); here we focus on protein variability. For monogenic disease, direct loss of drug efficacy has been reported due to drug target polymorphism. For eculizumab the protein haplotype forms of the C5 complement protein clearly impact its effectiveness in the Japanese population. It is likely that the immediate visibility of the effect of haplotype polymorphism (in a geographically focused population) on eculizumab effectiveness resulted from low disease heterogeneity in PNH and the high effectiveness of the drug. A-priori knowledge of the C5 protein haplotype distribution, as is now enabled by Haplosaurus, could potentially have been used to anticipate this lack of response.

C5 is one of only 32 gene targets of launched biologic drugs; another 189 are in human trials. For PNH, loss of eculizumab efficacy proved to be detectable, but in complex heterogeneous diseases, multiple genes act in combination with lifestyle and environmental factors. In large all-comer trials, common before the precision medicine era, protein haplotype diversity is only one aspect that may impact drug response efficacy and its direct effect is not always quantifiable or traceable. Despite the apparent risk paradox that suggests high haplotype diversity exists in these successful targets as analysed here, the data required to establish the true risk are simply inaccessible due to the disease heterogeneity in those earlier biologics trials. The number of protein targets for small molecule drugs is many folds greater than for biologics[37], and the real-world effect of protein haplotypes on small molecule efficacy remains an open question.

In our TLR4 example, we illustrated how protein haplotypes in populations can impact drug binding and activity and the importance of these may yet increase with more precise trials using patient sub-groups. That mAb-to-TLR4 activity falls inversely with the number of "deleterious-to-binding" haplotypes a patient carries highlights the importance of target zygosity and protein diplotypes. Although we can find no reported examples, one can conceptually extrapolate to cases of compound heterozygosity; where alleles of two different variants that each independently inhibit drug activity; in this case haplotype resolution would be required to predict drug activity according to whether an individual with both alleles carried them on the same (cis-

acting) or different (trans-acting) chromosomes of a pair. Further, recent work[38] has demonstrated a distinction between genes in abundance of cis-acting and trans-acting variants, reinforcing the importance of phase in interpretation of protein-coding genetic variation.

Sometimes target protein haplotype diversity occurs in regions outside the drug binding region and so are unlikely to have an impact. This assertion was used during prioritisation of protein haplotypes for FPR1 described earlier. A high-profile example recently reported elsewhere is PCSK9, the target of several cholesterol-lowering biologic drugs such as evolocumab. Although PCSK9 has seven common protein haplotypes, all protein-coding variants occur outside of neutralising mAb binding sites[39]. Regardless, we argue that most current and future drug discovery programmes would benefit from using Haplosaurus to inform the design of screening approaches that identify binders to multiple protein haplotype variants.

In our FPR1 example we showed how early consideration of protein haplotypes reduces risk of a drug development programme, but note that the 1% frequency threshold we used for protein haplotype consideration would have missed the C5:802V>I,885R>H haplotype responsible for loss of eculizumab efficacy had it been applied in that context. This raises the question as to whether the global 1% frequency threshold is generally appropriate? Precision medicine is not only about patient stratification, but also designing therapeutics against genes with complex target haplotype profiles such that they work across a broad population, for which sensitive analysis is crucial.

Although this work has focused on the impact of natural human protein haplotype diversity on mAb design, drug discovery must also consider sequence diversity during translational and toxicology studies where Haplosaurus could also be of value, particularly in out-bred primates. Beyond mAb design, we envisage a broad range of applications benefiting from convenient access to protein haplotypes that are now available via Ensembl because of Haplosaurus. Examples include those that require high target binding specificity such as the development of small molecule therapeutics or RNA and protein biomarker diagnostics, and it is also possible that haplotype analysis of somatic and germline mutations in tumour DNA may benefit design of cancer therapeutics. Other interested research communities include those studying protein structures and sequence evolution.

As an intermediate step to computing protein haplotypes, cDNA haplotypes are also generated. These are available for each alternatively spliced transcript (isoform) via Ensembl. This extends the scope of Haplosaurus for those studying cDNAs and the influence of both coding and non-coding variants on, for example, alternative splicing and transcript expression levels.

Haplosaurus makes it easy to prepare the input sequences for haplotype association studies both candidate gene and genome-wide[40,41]. In concert with extensive phenotype-to-genotype data, from initiatives such as The Cancer Genome Atlas[42], the 100,000 Genomes Project[43], and the Precision Medicine Initiative[44] and with improved resources such as the Haplotype Reference Consortium[20], our tools will continue to make gene haplotype analysis even more tractable and powerful over time.

## Methods

**Haplosaurus bioinformatics software**. The software consists of three main components (see Supplementary Fig. 4): (a) The Transcript Haplotype display on the Ensembl website. (b) The Haplosaurus command line tool. (c) Extensions to the Ensembl Perl application programming interface (API) used both by the Haplosaurus command line application and the Transcript Haplotype display. All three components have been adopted by the Ensembl project and will be actively maintained. Haplosaurus can be used with any of the 100 s of species for which there is an Ensembl database.

The Ensembl API[45] was extended to provide functionality for protein haplotype computation. The extensions have been integrated into the main API and may be used by those writing their own applications.

Transcript structure and sequence information is retrieved from Ensembl databases (see Supplementary Fig. 4). Phased genotypes are fetched from VCF files for the portion of genomic sequence bounded by the transcript's exons. Genotypes homozygous for the reference allele are filtered out, and monoploidy on male X and Y chromosomes is accounted for.

Genomic positions of genotypes are mapped relative to the CDS of the transcript. The allele pairs (if diploid) from each genotype are used to edit the reference CDS of the transcript at the mapped positions. Edits are applied in 3′ to 5′ order to avoid coordinate shifting caused by insertions or deletions. Each distinct CDS is translated in silico, creating a set of CDS Haplotype and Protein Haplotype objects.

Haplotypes are re-aligned with the respective reference sequence using a Miller–Myers global alignment method[46], as implemented in the BioPerl-ext package (https://github.com/bioperl/bioperl-ext). Contiguous blocks where the haplotype differs from the reference are stored as "diffs". Where possible, diffs are assigned identifiers of known variants. For single amino acid changes in protein haplotypes, pathogenicity scores from SIFT[47] and PolyPhen[48] are retrieved from pre-calculated matrices. Diffs are named by a notation similar to HGVS[49], and combined with the transcript or protein identifier to produce a haplotype identifier e.g., ENSP00000413079:29R>P,198G>S.

CDS Haplotype and Protein Haplotype objects are aggregated in a TranscriptHaplotypeContainer object. Haplotype counts across populations are used to derive frequencies as defined by a given panel or population structure. The container and constituent objects have hook methods for JSON serialisation.

The Haplosaurus command line tool (haplo) forms part of Ensembl's Variant Effect Predictor (VEP) toolset (https://github.com/Ensembl/ensembl-vep)[16]. It is designed to enable users to derive CDS and protein haplotype sequences for their own genotype data. Users input a VCF file containing phased genotypes for one or more individuals. Output consists of either a tab-delimited file or serialised JSON objects as above.

We have exposed aspects of the Haplosaurus API via a RESTful web interface (http://rest.ensembl.org/documentation/info/transcript_haplotypes_get). For a given transcript the REST API returns a JSON representation of a TranscriptHaplotypeContainer containing haplotypes derived from 1000 Genomes Project genotypes.

**Validation of Haplosaurus results**. Haplosaurus is applied to existing phased population variation data such as the 1000 Genomes data and testing was focussed not on the validity of this underlying data but on the validity of the methods used to create DNA and protein haplotypes from VCF files, genome reference sequence and annotations.

Haplosaurus was tested by comparison with independently developed software as well as by manual inspection. The validation procedure applied a comprehensive series of mutations (shown in Supplementary Table 1) to original gene sequences generating a simulated VCF file and two corresponding haplotype sequences per gene generated using independently developed software. We then used Haplosaurus to apply these simulated VCF files to the same original gene sequence and compared the resulting haplotype sequences to those generated above. Each VCF test passed if the comparison was identical or when, in a small number of cases, there were known differences in the treatment of variations in splice sites between the two methods. Test data are available (Supplementary Data 4[50]) and the results are shown in Supplementary Table 2.

The independent software that creates haplotype sequences from a VCF, DNA sequence and annotations was written for another project and has been contributed to the vcf tools package[51] here: https://github.com/vcftools/vcftools/blob/master/src/perl/vcf-haplotypes. In addition to generating haplotype sequences this programme also applies VCF changes to the original annotation GFF file to produce one new GFF file per haplotype each adjusted for any indels so that it refers to the correct regions in the corresponding altered sequence. Indels are applied even if they are near splice sites and the positions of annotations in the GFF file are adjusted without consideration of any resulting splice site changes. Haplosaurus does not apply any VCF changes in introns or in splice sites and this resulted in the small number of discrepancies between the two methods mentioned above. We concluded that the Haplosaurus behaviour was correct in all these cases.

**Protein haplotypes database from 1000 Genomes collection**. The haplotype database used to derive genome-wide statistics was created using an eHive pipeline[52]. The pipeline creates a job for each protein-coding transcript in the genome, each of which is then run in parallel on a distributed computing cluster.

The eHive pipeline invokes the Haplosaurus API as described in methods, serialising TranscriptHaplotypeContainer objects to simple tables for import into a MySQL database. Analyses presented in this paper were carried out directly on this database, or on exports from the database into the R environment.

The transcript database used was Ensembl version 83 (December 2015), human reference assembly version GRCh37. All 23,315 protein-coding genes from the primary assembly were included, representing 104,565 protein-coding transcripts.

Where we considered only a single transcript per gene, the canonical transcript was selected according to the following hierarchy: 1. longest consensus coding sequence (CCDS) translation with no stop codons; 2. longest Ensembl/Havana-merged translation with no stop codons; 3. longest translation with no stop codons. Although included in the database, our analyses exclude genes on ALT contigs, or where the canonical transcript for a protein coding gene was non-coding. The total number of genes analysed was thus 20,166.

Genotypes from the 1000 Genomes project VCF file (release 2014/07/30) were used to generate 2,220,113 protein haplotypes from 2504 individuals, with frequencies of occurrence (FoO) grouped into five continental super-populations. Limiting to those of the canonical transcripts yields 741,639 protein haplotypes. Genotypes had previously been phased according to the methods in[53]. The protein haplotype database for the 1000 Genomes Phase 3 dataset is available from http://www.figshare.com with [https://doi.org/10.6084/m9.figshare.5545084].

For comparison with the UniProt and RefSeq Protein sequences we used Haplosaurus to generate a file of each protein haplotype sequence from all 2,220,113 entries (Supplementary Data 5[54]). We downloaded the 71,579 Human Proteome sequences of UP000005640 from July 18, 2017 from the UniProt website (http://www.uniprot.org/) and the 45,084 RefSeq human proteins corresponding to GRCh37 from Release 84 from the NCBI website (https://www.ncbi.nlm.nih.gov/projects/genome/guide/human/). Exact mapping of the full-length UniProt and RefSeq sequences to full-length protein haplotypes, minus any trailing stop codon, was performed using a simple Perl script (Supplementary Software 2[55]). Results were loaded into the MySQL database for analysis [https://doi.org/10.6084/m9.figshare.5545084]. 62,203 UniProt proteins mapped to a protein haplotype, 9375 proteins did not. 38,186 RefSeq proteins mapped to a protein haplotype, 6897 did not. Considering just canonical transcripts, 3193 genes had no mapping to any UniProt entry and 1646 genes had no mapping to RefSeq, even taking in to account protein haplotypes. In many cases the failure to map could be accounted for by differences in gene models of the manually curated UniProt and RefSeq entries vs. automated Ensembl predictions rather than natural protein variation.

In our analyses, we use a threshold of 1% FoO above which we assume protein haplotypes are of significance to drug development. The sensitivity of this threshold on the number of protein haplotypes flagged as significant, and the number of significant protein haplotypes per gene, is shown in Supplementary Fig. 5. At a FoO threshold of 1%, 55,108 of the 741,639 haplotypes (canonical transcripts only) are significant; this seemingly low fraction is balanced by the observation that well over half (432,360) haplotypes are only seen once in the 1000 Genomes dataset. Reducing the FoO threshold fivefold from 1 to 0.2%, a level at which the C5:802V>I,885R>H haplotype responsible for loss of eculizumab efficacy would have become significant, results in a doubling of the overall number of significant haplotypes to 109,808. Increasing the FoO threshold fivefold from 1 to 5% reduces their number by a third, to 34,402. We have asserted that the number of significant haplotypes per gene is an important consideration in drug development. At a FoO threshold of 1%, 7708 genes have a single haplotype, 7671 have two or three and 4785 have four or more. Reducing the threshold to 0.2% increases the number of genes with four or more (those genes for which target protein haplotype prioritisation in drug development may be required) to 10,173; i.e., over half of all genes. We also recognise that in drug development a key metric is the number of patients who could be impacted by haplotype variability: this depends not only on the number of haplotypes but the relative frequencies of them. If we consider increasing the FoO threshold from 1 to 5%, the number of genes with a single haplotype increases from 7708 to 11,444. In other words, 7708 genes have a single dominant haplotype that covers over 98% of the population (allowing for heterozygosity) whilst 11,444 genes have a single haplotype that covers over 90%.

**Ensembl transcript haplotype displays powered by Haplosaurus**. When considering a single gene, as we did for C5, TLR4 and FPR1 we drew data from the Ensembl Transcript Haplotype display, which is an integral component of the main Ensembl website and powered by the Haplosaurus API. The web display contains more detail on a per-gene level, including haplotype frequencies for all 26 of the 1000 Genomes populations (the genome wide database included frequencies only for the five 1000 Genomes super populations) and protein haplotype zygosity (not included in the database). In this section, we show the use of the Transcript Haplotype display with reference to the C5 example described in the main paper (Supplementary Fig. 1). Being part of the main Ensembl website means that the underlying data will be updated with each Ensembl release.

**Variants vs. haplotypes in analysis for Fig. 2**. The objective was to see how complex the relationship between individual protein variations and protein haplotypes is.

The analysis was performed using data from the 1000 Genomes haplotype database described in this paper. As explained above the database provides a table of protein haplotypes where for each haplotype a separate table contains related entries describing each difference to reference ("diff") observed in that haplotype. A diff is uniquely defined by the protein in which it occurs, the starting position and the change to the reference. The term "variation" is synonymous with "diff" and is used throughout the manuscript and hereafter in this section. Each position can have multiple variations.

The R programming language (https://www.r-project.org)[56] and RStudio (https://www.rstudio.com)[57] were used to perform further analysis of the database. The R script used to produce Fig. 2 is available via Supplementary Software 1[25].

The FoO for each variation was calculated from the frequency of all the haplotypes containing it. For each protein, the list of "significant" variations with FoO ≥ 1% was determined. The number of variations in this list for each protein was used on the x-axis for Fig. 2. The y-axis is effectively the minimum number of highest frequency haplotypes required to include all these significant variations, this is explained further below.

A variation in a haplotype cannot itself have an overall frequency of less than the haplotype, but it can be higher if it occurs in other haplotypes too. Our hypothesis was that the same variation could be found in many haplotypes accompanied by many other polymorphic positions such that the variation itself has overall FoO >= 1% but it occurs in no haplotypes with FoO >= 1%. Such a variation would be missed if relying wholly on haplotype frequency for inclusion. We tested this hypothesis and found 1449 canonical proteins with at least one variation for which this is the case. Looking only at the highest frequency variation for each protein the frequencies of these "diluted" variations range from 1.02% to 97.3% with a median of 1.32%, mean of 2.16%, 1st quartile 1.12% and 3rd quartile 1.74%. Therefore, most of these diluted variations have a low frequency with some exceptions such as in protein MUC4 which has 12 variations that occur at FoO >= 50% but are not found in any haplotypes with FoO >= 1%. To account for these diluted variants the y-axis in Fig. 2 counts haplotypes with FoO >= 1% and adds to those the minimum number of haplotypes of the highest frequency to include any remaining variants with FoO >= 1%. We will investigate ways to account for these diluted variations in future versions of Haplosaurus.

**Measuring the biological activity of mAbs for TLR4.** PBMC isolation: Human peripheral blood mononuclear cells (PBMCs) were isolated from leucocyte cones (supplied by NHS Blood and Transplant Service (NHSBT, UK) as anonymized samples from consenting donors).

Antibodies: An IgG1 mutant lacking effector function was used[58] to avoid possible complications involving effector function in cell-binding antibodies. TLR40090 IgG1-TM (batch IG200809-02, stock concentration 0.41 mg/ml) and CAT254 IgG1-TM (isotype control) (batch SP09-035, stock concentration 13.98 ml/ml) were used in the experiments shown.

LPS-induced TNFα production (Supplementary Fig. 2a): TNFα was released in the supernatants of PBMC after 24 h incubation with LPS (derived from *Salmonella minnesota*, Calbiochem) in the presence or absence of inhibitory and isotype control antibodies. TNFα concentration was measured by ELISA (TNFα Duoset, R&D Systems). The final detection step was performed using streptavidin–Europium conjugate and standard DELFIA reagents (Perkin Elmer).

Flow cytometry with transiently transfected cell lines (Supplementary Fig. 2b): Variants of human TLR4 in pUNO vector were transiently transfected into HEK 293 cells using Lipofectamine 2000 FACS analysis was carried out 72 h post transfection using TLR40090 IgGTM-PE, 15C1 IgGTM-PE[59] or CAT254 IgGTM-PE isotype control. Direct labelling with phycoerythrin was performed using Zenon R-Phycoerythrin Human IgG Labelling Kit (ThermoFisher).

**Measuring the biological activity of mAbs for FPR1.** FPR1 antibody FPR0155 and control IgG production: High affinity antibodies to FPR1 were identified as previously described[35].

Cell lines: FPR1 reporter cell lines, comprising CHO cells transfected with each of the human FPR1 variants in combination with the human G-protein subunit Gα16, were used for flow cytometry and to identify antibodies that were able to inhibit the activation of FPR1 by formyl peptides. Some differences in FPR1 expression levels between FPR1 variant cell lines were noted.

Flow cytometry with stably transfected cell lines (Supplementary Fig. 3a): FACS analysis was carried out using Fpro0155 IgG with detection using a PE-labelled anti- human IgG (Sigma P8047). Isotype controls were in house human IgG1 antibody. Formyl peptide induced calcium signalling assays (Supplementary Fig. 3b): Intracellular calcium release upon FPR1 stimulation with formyl peptides was measured using a calcium-sensitive fluorophore (FLUO-4 NW Calcium Assay kit (Molecular Probes)) in a plate-based fluorescence detection system (FLIPR-tetra, Molecular Devices). Antibody titrations were added to human FPR1 cells in presence of the calcium-sensitive FLUO-4 dye and probenecid, Fluorescence was measured for a period of 3 min after addition of formyl peptide (fMLFF, Bachem) and peak $Ca^{2+}$ signal and percentage maximal response were derived from the data.

**Code availability**. Haplosaurus is part of the Ensembl project. All source code and installation instructions are freely available under an Apache 2.0 license from GitHub: http://github.com/Ensembl.A snapshot of the Ensembl code used to generate the results published in this manuscript are available via [https://doi.org/10.6084/m9.figshare.6834068.v1], [https://doi.org/10.6084/m9.figshare.6834071.v1] [https://doi.org/10.6084/m9.figshare.6834074.v1].Supplementary Software 1: The R scripts used to generate Fig. 2 are available via [https://doi.org/10.6084/m9.figshare.6834008.v1].Supplementary Software 2: The Perl script used for mapping full-length UniProt and RefSeq sequences to full-length protein haplotypes is

available via DOI [https://doi.org/10.6084/m9.figshare.6834248.v1].All code available via the preceding DOIs is freely available under Apache 2.0 licenses.

## Data availability

The protein haplotype database we built from the 1000 Genomes Phase 3 dataset is available via [https://doi.org/10.6084/m9.figshare.5545084]. Supplementary Data 4: Data used to test/validate Haplosaurus. Data are available via [https://doi.org/10.6084/m9.figshare.6834083.v1]. Supplementary Data 5: Fasta protein sequences of all protein haplotypes in 1000 Genomes. Data are available via [https://doi.org/10.6084/m9.figshare.6834191.v1].

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

## Acknowledgements

This project resulted from a grassroots collaboration between MedImmune (a global pharmaceutical company), Eagle Genomics (a bioinformatics software company) working under contract to MedImmune, and EMBL-EBI, with the latter funded by the Wellcome Trust (grant number108749/Z/15/Z) and the European Molecular Biology Laboratory. We would like to thank O.M. Harari and A. Gevorgyan for assistance with Haplosaurus test data and scripts; B. Ailey and C. Baloglu for assistance with early prototypes; and R. Minter, T. Wilkinson and M. Woodwark for assistance and/or helpful discussions; J. Ferguson and J. Large for technical assistance.

## Author contributions

T.S., C.M.K. and R.B. conceived the Haplosaurus concept. W.S. developed the initial Haplosaurus prototype. W.M. developed the final Haplosaurus implementation. W.M. and W.S. built the protein haplotype database. T.S. and W.S. performed the analysis of the protein haplotype database. C.C.H., L.E., D.R., J.C. M.J.R. and D.K.F. performed experiments related to, or led the TLR4 or FPR1 drug discovery projects for which the analyses are described. C.C.H. and T.S. performed review of marketed biopharmaceuticals, and D.K.F. interpreted this in a Precision Medicine context. C.C.H. and W.S. coordinated the manuscript. C.C.H., W.S., T.S., W.M., D.K.F. and F.C. wrote the paper. T.V., R.B., C.H., P.F. and F.C. supervised the project.

## Additional information

**Competing interests:** T.S., D.F., R.B., J.C., L.E., D.R., C.M.K., M.J.R., C.H., T.V. and C.C.H. are or were employees of MedImmune and may have received AstraZeneca shares as part of their usual compensation. WS has received shares in Eagle Genomics Ltd. PF is a member of the scientific advisory boards for Fabric Genomics, Inc., and Eagle Genomics, Ltd. The remaining authors declare no competing interests.

