## [Peer Review File · Nature Communications]

Reviewer #1 (Remarks to the Author):

Review of Spooner et al: "Haplosaurus: Computation of Protein Haplotypes for use in Precision Drug Design"

The authors describe a tool, "Haplosaurus," for aggregating haplotype information encompassing amino acid sequence variant combinations that might be of functional significance for encoded proteins. The utility of the tool is exemplified in three brief case studies in which knowledge of the population frequencies, or biological implications, of different amino acid sequences associated with a single protein led to insights about the utility of particular therapeutics that target that protein. Overall, I found the paper concise and insightful. I also believe that phase information is often ignored in genomic studies and this paper provides compelling examples of why this should not continue. I do have a couple of concerns/questions that the authors should consider. I outline these below.

1. What about alternative splicing and the frequency of splice variants associated with a particular gene? If there is a mutation that deforms a protein in a functionally significant and therapeutically relevant way but it is in an exon that is often spliced out, what might that mean in the grand scheme of things? Are there useful examples where sensitivity to the frequency of haplotypes associated specifically with different splice variants makes sense?
2. What about homozygosity vs. heterozygosity? There are many instances where a wild type copy of a gene/protein haplotype compensates for a mutant homologous copy of that gene/protein. Therefore, the frequency of homozygosity, not just haplotype frequency, is likely to be important.
3. The authors are correct to point out that most computational or population frequency-based phasing programs work reasonably well, but this is on a whole genome basis and is really biased toward accurately phasing common variants, as they are more likely to have reliable haplotype frequency information for imputation purposes. Unfortunately, phasing rare variants can be problematic if population-frequency imputation-based strategies are used. A discussion of this is important.
4. Tumor DNA is evaluated often to determine potential oncogenic mutations and drug targets. It is known that in certain instances precision drugs will not work if certain mutations are present (e.g., as in the case of BRAF inhibitors; see Rosen et al: PMID: 28783719). Exploring tumor haplotypes might be of interest, especially those that include both coding germline variants and somatic mutations, although phasing tumor DNA may be an issue.
5. The focus is on coding variants for obvious reasons, but non-coding variants can influence, e.g., splicing and transcript expression levels, which can both impact protein function. Mention of this would be appropriate.

6. In the context of haplotype frequencies, the combination of homologous gene haplotypes one possesses impacts molecular physiology, not each haplotype independently. For example, it may be the case that an individual does indeed have a combination of amino acid substitutions on one haplotype that are of consequence for the encoded protein, but the other copy of the gene is, e.g., deleted or damaged in some other way. In this setting, the presence of the non-functional copy of the gene “brought out” the deleterious effect of the amino acid combination on the other haplotype, which may not have occurred if the other haplotype was wild-type and did not result in haploinsufficiency. Thus, “compound heterozygosity” of this sort could be as important as combinations of cis-acting variants.

Reviewer #2 (Remarks to the Author):

In their article “Haplosaurus: Computation of Protein Haplotypes for use in Precision Drug Design”, Huntington and colleagues (i) report a “first-in-class” bioinformatics tool for computing protein haplotypes from pre-existing genomic variation data, and its integration into the Ensembl website; (ii) apply Haplosaurus to build a database of protein haplotypes from the 1000 Genomes dataset and use this database for the analysis of “the landscape of protein haplotype variability” “on a previously unexplored scale”; and (iii) demonstrate the potential impact of Haplosaurus on drug discovery through three case studies. Taken together, this article appears to represent the combination of a bioinformatics tool description, a resource paper, and a research report type of paper. Are the advances made by the development of the bioinformatics tool Haplosaurus (as suggested by the title), or by any of these components/parts in themselves, or in combination, sufficient to merit publication in Nature Communications? This question cannot yet be decisively answered, due to shortcomings in the current version of the manuscript which make it difficult to evaluate the conceptual and concrete advances. These concern (i) the presentation of the bioinformatics tool in the main text, which appears insufficient, that is, does not give the reader (who is not intrinsically familiar with the issues) a real idea of the achievements over previously available approaches and tools (while the information in the Supplementary Information is much more helpful); (ii) the integration of the work into existing literature; (iii) its conceptual foundation, and (iv) the clarity of presentation, in particular the lack of clear explanation and definitions. The authors will need to prepare a more homogeneous, coherent outline, i.e. ‘the story’ of the development of the tool and its - scientifically validated - application to the generation of a resource of value for precision drug design. They will have to pay more attention to consistency in detail and avoid redundancies. Notwithstanding these concerns, the population-based dissection of (diploid) gene/protein sequences into their protein haplotypes is a valuable contribution to the emerging field of diploid genomics (Tewhey et al., Nat Rev Genet 2011; 12:215-223) and expands on recent groundwork on the haplotype/diplotype architecture of diploid human genomes (Hoehe et al., Nat Commun 2014; 5:5569) in scale, ease of access and use for analysis, and concrete applicability to a medically relevant subject.

To (i), the presentation of the bioinformatics tool Haplosaurus in main text: this is more or less based on reference to Figure 1, steps d-h, essentially the in-silico translation of CDS haplotypes resulting in protein haplotypes, which have been phased in an earlier step by any established statistical phasing method, and the subsequent generation of population frequency data for the protein haplotypes. For further comments in this context see also 'Specific comments' below. The 'impact' of Haplosaurus, i.e. its benefits compared to the current practice of protein haplotype computation (in the area of drug development) as shown in Figure 1 are inseparable from the availability of 1000 Genomes data – so what is the accomplishment of this development in itself? In the Results section, information on Haplosaurus is disproportionately short (few lines).

To (ii), the integration of this work into existing literature: In contrast to the most widely used 'haplotypes', the specific term 'protein haplotypes' has not yet been established and only very rarely been used. It should at some point also be defined relative to the more popular term 'protein isoforms'. Notably, a first systematic, population-based analysis and description of 'protein haplotypes' and their 'frequencies of occurrence' (FoO) has been performed by Hoehe et al. (Nat Commun 2014;5:5569); in this work, protein haplotypes in European populations have been quantitatively assessed in several aspects (f.i. average numbers of different protein haplotypes per gene; the increase of protein haplotype numbers as a function of genome numbers; the fractions of different, unique protein haplotypes relative to total haplotype counts in increasing numbers of genomes), and genes have been classified into different categories based on FoO of their unique protein haplotypes. This work, which laid groundwork for gene and protein haplotype (and diplotype) architecture of human genomes, should be acknowledged appropriately in all its relevant aspects. It is inadequate to cite this work only by referring to the high concordance between statistically and molecularly generated haplotypes it shows (and the specific citation of this aspect is 'not exactly right'), serving as a justification for Haplosaurus. For another protein haplotype-based work utilizing statistically and molecularly phased genomes see <https://www.biorxiv.org/content/biorxiv/early/2017/11/17/221085.full.pdf>

To (iii) and (iv), the conceptual foundation of this work and necessary explanations and definitions (in variable order): The basic terms need to be defined/explained more clearly, and concepts made explicit. The general definition of 'haplotype' in the Introduction ("Such differences result from the unique combination of DNA variants (genotype) in each individual, inherited together from a single parent") sounds awkward, needs to be phrased more clearly and checked against definitions established in the literature. Then, the (commonly used) genetic definition of haplotypes as the combinations of variants on each of the two chromosomes (of a gene) is effectively not compatible with the analysis of protein haplotypes defined as the translated transcripts of gene haplotypes (a molecular definition of haplotypes independent of existing variants; BTW the term 'haploid chromosome' is inappropriate!); these also include invariable haplotype sequences identical with the reference sequence. It should furthermore be stated explicitly that the 'protein haplotypes' mean 'different', or 'unique' haplotypes (which include the reference haplotype), and these need to be distinguished from the entirety of haplotypes in a population, the 'total haplotype count', equivalent to the total number of genes x 2 x number of genomes/individuals. The frequency of occurrence (FoO) of a protein haplotype of a gene needs to be defined, supposedly calculated as: counts of a unique haplotype of a gene in a population, divided by total haplotype count of this gene, equivalent

to 2x number of individuals/genomes = $2 \times 2504 = 5008$. The definitions of all terms defining the overall approach should be explicitly formulated in main text.

Also, it has remained unclear to me, whether a priori only genes with mutations annotated by Ensembl's Variant Effect Predictor are included in analysis, or all genes with amino acid exchanges, and to which extent/at which stage, genes with functionally protein-altering mutations predicted by PolyPhen-2 and SIFT are included. More information on the choice and use of mutation algorithms is needed. Finally, given that a certain fraction of genes are invariable, it is not clear whether the average numbers of protein haplotypes per gene are calculated across all (variable and invariable) genes, or across variable genes only.

The rationale underlying the selection of the most frequent protein haplotypes in the population as the basis for individualized drug design needs to be outlined more coherently at the beginning. If 'frequency of occurrence' of a protein haplotype guides the selection, this apparently may work for drugs, which do not target the specific causative, disease-related protein, which presumably is rare in the population, and effectively requires knowledge of the genetic risk profile. This key point has been addressed in examples, bits and pieces all over the manuscript, at best probably in discussion. It should be presented, however, comprehensibly in the Introduction.

In diploid genomes, gene/protein functions are mediated by two forms of the gene/protein, i.e., by pairs of haplotypes (diplotypes). The specific role of each of these two haplotypes remains to be largely clarified, as does their (differential) expression status. Drysdale et al. (PNAS 2000; 97: 10483-10488) have, for example, demonstrated the importance of diplotypes over haplotypes in relation to drug response. These issues and their impact on the authors' approach have not been addressed at all.

Specific comments:

The Introduction needs substantial revision and should be much more clearly developed, avoiding redundancies.

First section: Comments on definitions see above; second half partially redundant to Abstract.

Figure 1: Steps d) and e) remain unclear to me. Why is the reference transcript model not presented as the two haplotypes derived earlier (Phased genotypes in c)? "Sequence editing": the whole sentence appears unclear; this step should possibly be renamed and reformulated; "one for each phase": superfluous.

Step g): The arrow should point towards population analysis and from there to the box with haplotype frequencies.

The illustration of 2^n haplotypes should be omitted; it is generally understood that the theoretically expected number of haplotypes is not observed, i.e. that the observed number of haplotypes is much smaller due to LD.

Chapter "Creating a protein haplotype collection...":

"The total number of protein haplotypes $F_{O \geq 1\%}$ in our database is 55,108...": even though the total numbers given in the Results sections look very large, they should be put in perspective: 55,108 protein haplotypes with $F_{O \geq 1\%}$ are different among a total haplotype count of $(20166 \text{ genes} \times 2 \times 2504) = 100,991,328$; thus, the fraction of different protein haplotypes among the total number of existing haplotypes is equivalent to $\sim 0.0005\%$, the remaining haplotypes are multiples of these different haplotypes - a dramatic reduction compared to the numbers of different gene haplotypes.

Reviewer #3 (Remarks to the Author):

This manuscript describes an interesting approach to provide a repository of information about human protein sequences, in a way that properly reflects real-life variation in the sequence. The database largely draws its information on variation from the 1000 Genomes Project data, and reconstructs the variant haplotypes that would be predicted in the polypeptide sequence. This is expected to be useful in the design of therapeutics (either monoclonal antibodies or drugs) by defining the range of commonly encountered protein variants that would be expected in the population.

I thought that the idea was a sensible one, and in particular it is important that the authors recorded composite haplotype combinations rather than simply documenting missense variation at individual sites. Although it is not particularly surprising that the reference sequence for as many as one in seven genes is not the commonest – indeed I would have expected rather more – this is a simple illustration of the usefulness of the resource in predicting the best sequences to test with a drug or antibody within or across populations. The examples shown illustrate the potential usefulness, though the added value of analysis by haplotype is not obvious (compared with analysis of individual amino acids).

In general I thought the report was clearly written, and I thought it was useful that HGVS nomenclature had been adopted. There are nevertheless some aspects of the detailed presentation of this work that need attention. In particular, those already familiar with human genetic variation

will find some of the home-made innovations of terminology distracting and/or misleading. I do not understand why the authors persistently use the strange acronym FoO, when the perfectly good word "frequency" already exists. Similarly (line 127), the word "significant" should be reserved for statistically significant observations, and its meaning should not be changed to indicate classification by frequency. Overall, it will detract from the clarity and reach of the paper to describe as significant or FoO >1% haplotypes which would conventionally be simply described as "MAF >0.01".

The first sentence of the Introduction (lines 27-28) declares that there are human protein haplotypes that differ at hundreds of amino acid positions. I find it very hard to believe that this is literally true, and I would be interested to see a single example. Looking at the genes (Supplementary Table 1) listed as having the largest number of variant haplotypes, many of the highest-scoring genes (like FLG, ANHAK2 and the mucin genes MUC4, 2, 16, 12 etc.) are famous for containing variably repeated coding sequences, and I think it is very likely that the extreme apparent divergences between haplotypes in these genes are artefacts of misalignment rather than truly high frequencies of missense (substitutional) variation.

Finally, it is noteworthy that there is almost no description of methodology in the paper itself, with the description of how Haplosaurus was put together relegated to Supplementary Notes. I think that at the very least there should be an outline description of how phased 1000 Genomes Project data were integrated into the study. The Supplementary Note implies that phased haplotypes were directly imported from the 1000 Genomes Project; that is likely to be reliable for most genes, but I found it difficult to imagine how the correct protein haplotypes could be reconstructed from doubly heterozygous individuals for genes that crossed recombination hotspots. Presumably in such cases there will be free association between different LD blocks, but I think this is a non-trivial question that deserves at least some comment in the main paper.

Author's response to Reviewers' comments

Dear Editor,

We have addressed each of the specific comments raised by the three reviewers. Each comment, and how we have incorporated changes into the revised manuscript, are described below

Response to Reviewer #1

We thank the Reviewer for their thoughtful comments, and agree that the importance of phasing information is often underappreciated in many studies. Specific comments are addressed below.

Comment 1. "What about alternative splicing and the frequency of splice variants associated with a particular gene? If there is a mutation that deforms a protein in a functionally significant and therapeutically relevant way but it is in an exon that is often spliced out, what might that mean in the grand scheme of things? Are there useful examples where sensitivity to the frequency of haplotypes associated specifically with different splice variants makes sense?"

A1. Although the type of analysis the Reviewer suggest could be performed, we have not done so for two main reasons. a) Simplicity; the importance of protein haplotypes is generally underappreciated by the protein science community even at the level of the canonical isoform and to introduce complexities of alternative splicing may distract from the central message of the paper. b) Lack of a case study; whilst the situation that the Reviewer describes is highly plausible we have not seen it reported in the literature or come across it in our own work.

We now state in paragraph 1 of the section titled "Creating a protein haplotype collection from the 1000 Genomes dataset" that: "Although our database contains protein haplotypes for each alternatively spliced transcript (isoform) annotated by Ensembl we have, for simplicity, restricted our analysis here to a single canonical isoform for each gene" (page 5, lines 26-28).

That said, the issue of alternative splicing/isoforms is an important one, and we have further clarified the potential for application of our tools in this regard in the discussion: "As an intermediate step to computing protein haplotypes, cDNA haplotypes are also generated. These are available for each alternatively spliced transcript (isoform) via Ensembl. This extends the scope of Haplosaurus for those studying cDNAs and the influence of both coding and non-coding variants on, for example, alternative splicing and transcript expression levels"(Page 14, lines 21-25).

Comment 2. "What about homozygosity vs. heterozygosity? There are many instances where a wild type copy of a gene/protein haplotype compensates for a mutant homologous copy of that gene/protein. Therefore, the frequency of homozygosity, not just haplotype frequency, is likely to be important."

A2. We agree that the issue of zygosity (i.e. diplotypes) was not sufficiently covered. We have addressed this with the following additions:

- in the reworking of our introduction, through consideration of diplotypes; "In diploid genomes, function is mediated by two forms of the gene/protein, i.e., by pairs of haplotypes (diplotypes)"(page 1, lines 39-41).

- in the description of the database through a count of the number of common diplotypes *“In our database of 2,504 genomes across 20,166 genes (a total of 100,991,328 haplotype observations and 50,495,664 diplotype observations) we observe a total of 718,964 unique protein haplotypes and 1,068,742 unique protein diplotypes”* (page 6, lines 2-4).
- in the description of TLR4 example *“Analysing diplotype distributions we see, in the full 1000 Genomes set, 12.1% of individuals have at least one 299D>G haplotype copy (reduced activity of MEDI-2843) and 0.4% have two (complete loss of MEDI-2843 activity)... This example demonstrates that assessing not only the overall protein haplotype frequency but also their diplotypes, as enabled by Haplosaurus, can provide important insight.”*(page 10, lines 23-33).
- and in the discussion where we point out that the diplotype, not just haplotype is a crucial factor in drug response: *“That mAb-to-TLR4 activity falls inversely with the number of “deleterious-to-binding” haplotypes a patient carries highlights the importance of target zygosity and protein diplotypes. Although we can find no reported examples, one can conceptually extrapolate to cases of compound heterozygosity; where alleles of two different variants that each independently inhibit drug activity; in this case haplotype resolution would be required to predict drug activity according to whether an individual with both alleles carried them on the same (cis-acting) or different (trans-acting) chromosomes of a pair”*(page 13, lines 31-37).

Comment 3. *“The authors are correct to point out that most computational or population frequency-based phasing programs work reasonably well, but this is on a whole genome basis and is really biased toward accurately phasing common variants, as they are more likely to have reliable haplotype frequency information for imputation purposes. Unfortunately, phasing rare variants can be problematic if population-frequency imputation-based strategies are used. A discussion of this is important.”*

A3. We agree with the reviewer. In response, we have clarified the part of the introduction where we discuss phasing techniques as follows: *“A criticism of imputation-based phasing methods such as those used for 1000 Genomes [19] is that they are ineffective for rare and de novo variants [20]. However, in drug discovery it is the common haplotypes that are most significant, and these are more likely to be based on reliable variant frequency information needed for imputation.”* (page 4, lines 3-7)

Comment 4. *“Tumor DNA is evaluated often to determine potential oncogenic mutations and drug targets. It is known that in certain instances precision drugs will not work if certain mutations are present (e.g., as in the case of BRAF inhibitors; see Rosen et al: PMID: 28783719). Exploring tumor haplotypes might be of interest, especially those that include both coding germline variants and somatic mutations, although phasing tumor DNA may be an issue”.*

A4. We agree that the protein haplotypes of tumour genomes (which our tools can readily be used to compute) are likely to be of great interest in drug discovery, and now mention this in the discussion to this effect *“it is also possible that haplotype analysis of somatic and germline mutations in tumour DNA may benefit design of cancer therapeutics”* (page 14, lines 18-19).

Comment 5. *“The focus is on coding variants for obvious reasons, but non-coding variants can influence, e.g., splicing and transcript expression levels, which can both impact protein function. Mention of this would be appropriate.”*

A5. We agree that haplotypes involving non-protein coding variants are important. Our tools already generate cDNA haplotypes, and we have added a note to that effect in the discussion: *“As an intermediate step to computing protein haplotypes, cDNA haplotypes are also generated. These are available for each alternatively spliced transcript (isoform) via Ensembl. This extends the scope of HaploSaurus for those studying cDNAs and the influence of both coding and non-coding variants on, for example, alternative splicing and transcript expression levels.”* (page 14, lines 21-25).

Comment 6. *“In the context of haplotype frequencies, the combination of homologous gene haplotypes one possesses impacts molecular physiology, not each haplotype independently. For example, it may be the case that an individual does indeed have a combination of amino acid substitutions on one haplotype that are of consequence for the encoded protein, but the other copy of the gene is, e.g., deleted or damaged in some other way. In this setting, the presence of the non-functional copy of the gene “brought out” the deleterious effect of the amino acid combination on the other haplotype, which may not have occurred if the other haplotype was wild-type and did not result in haploinsufficiency. Thus, “compound heterozygosity” of this sort could be as important as combinations of cis-acting variants.”*

A6. Accounting for compound heterozygosity is a great example of added value haplotype analysis vs. analysis of individual amino acids. We have brought this out in our discussion of the TLR4 case as follows: *“That mAb-to-TLR4 activity falls inversely with the number of “deleterious-to-binding” haplotypes a patient carries highlights the importance of target zygosity and protein diplotypes. Although we can find no reported examples, one can conceptually extrapolate to cases of compound heterozygosity; where alleles of two different variants that each independently inhibit drug activity; in this case haplotype resolution would be required to predict drug activity according to whether an individual with both alleles carried them on the same (cis-acting) or different (trans-acting) chromosomes of a pair.”*(page 13, lines 31-37).

Response to Reviewer #2

We appreciate that our text, which was written from a Biotechnology perspective, may not always be appropriate for a specialist genetics/genomics audience. We have taken on board the Reviewer's comments to comprehensively rework parts of the introduction and results and we address specific comments below.

***Comment 1.** To (i), the presentation of the bioinformatics tool Haplosaurus in main text: this is more or less based on reference to Figure 1, steps d-h, essentially the in-silico translation of CDS haplotypes resulting in protein haplotypes, which have been phased in an earlier step by any established statistical phasing method, and the subsequent generation of population frequency data for the protein haplotypes. For further comments in this context see also 'Specific comments' below. The 'impact' of Haplosaurus, i.e. its benefits compared to the current practice of protein haplotype computation (in the area of drug development) as shown in Figure 1 are inseparable from the availability of 1000 Genomes data – so what is the accomplishment of this development in itself?*

A1. The Reviewer appears unclear on the accomplishment of the development of Haplosaurus and that it is *"inseparable from the availability of 1000 Genomes data"*. A useful analogy is the development of tools to call variants from sequence alignment files; it is quite feasible to call variants by manual inspection of the alignments in a region of interest, or to create bespoke scripts to call alignments for a specific application. However, variant callers (cf Haplosaurus in the context of protein haplotype computation) make the process much more efficient, reliable and reproducible. We should also note that, before Haplosaurus *"we have found no methods that can compute protein haplotype data from phased genotypes, or resources that make precomputed protein haplotypes available"* (page 4, lines15-16). We hope that our extended description of the Haplosaurus tool will clarify its accomplishments (page 4, lines 40-46 and page 5, lines 1-18).

***Comment 2.** In the Results section, information on Haplosaurus is disproportionately short (few lines).*

A2. We agree, and have reworked text from the supplementary information into the Results section of the main manuscript: *"For a given gene identifier and VCF file, Haplosaurus retrieves the corresponding gene model from a linked Ensembl database. Next, the phased variants that overlap the gene's location are retrieved from the VCF file and used to generate two lists of DNA sequence alleles for each sample, one for each DNA haplotype, according to their phase. The DNA haplotype sequences are reconstructed by substituting the alleles for each haplotype into the reference sequence according to their genomic location. The DNA haplotypes are virtually 'transcribed' to coding sequence (CDS) haplotypes, and these are then virtually 'translated' into protein haplotypes. Once the protein haplotypes for many samples have been generated, the software can calculate the FoO of each unique protein haplotype sequence in a population."* (page 5, lines3-12).

***Comment 3.** In contrast to the most widely used 'haplotypes', the specific term 'protein haplotypes' has not yet been established and only very rarely been used. It should at some point also be defined relative to the more popular term 'protein isoforms'.*

A3. We agree that the term “protein haplotypes” is not often used but anticipate that this may change as a consequence of this and similar works. Clear distinction between protein haplotypes and protein isoforms has been addressed in the first paragraph of the Introduction: *“Proteoforms are the different molecular forms in which the protein product of a single gene can be found [1]. Proteoforms modulate a wide variety of biological processes and contribute to many phenotypes and diseases [2]. There are two main classes of proteoforms; protein isoforms, which include alternatively spliced RNA transcripts and post-translational modifications, and protein haplotypes, where protein changes are due to genomic variation.”* (page 1, lines 26-30).

Comment 4. *Notably, a first systematic, population-based analysis and description of ‘protein haplotypes’ and their ‘frequencies of occurrence’ (FoO) has been performed by Hoehe et al. (Nat Commun 2014;5:5569); in this work, protein haplotypes in European populations have been quantitatively assessed in several aspects (f.i. average numbers of different protein haplotypes per gene; the increase of protein haplotype numbers as a function of genome numbers; the fractions of different, unique protein haplotypes relative to total haplotype counts in increasing numbers of genomes), and genes have been classified into different categories based on FoO of their unique protein haplotypes. This work, which laid groundwork for gene and protein haplotype (and diplotype) architecture of human genomes, should be acknowledged appropriately in all its relevant aspects.*

A4. We have further acknowledged the work of Hohe *et al.* as follows;

- *“The groundwork for protein haplotype and diplotype architecture of human genomes was laid by Hoehe et al [8] who described a systematic, quantitative, population-based analysis of protein haplotypes in European populations.”* (page 1, lines 43-45).
- *“the frequency at which each unique protein haplotype occurs within a population is an important measure of potential clinical impact. We use frequency of occurrence, FoO (following the terminology of Hoehe et al [8])”* (page 2, lines 19-21).
- *“The entirety of unique haplotypes/diplotypes increases in relation to increasing number of genomes [8].”* (page 6, lines 1-2).
- *“Following Hoehe et al[8] we classify genes into distinct categories based on their number of common haplotypes. This provides an indication of protein haplotype complexity, which is a consideration in drug design.”* (page 6, lines 11-13)
- A direct comparison with Hoehe *et al* on “number of protein haplotypes per gene” was not possible due to the different genome numbers/frequency thresholds used.

Comment 5. *For another protein haplotype-based work utilizing statistically and molecularly phased genomes see [https link provided]*

A5. We thank the reviewer for highlighting this recent preprint. We have now referenced it in the section on phase in the discussion. *“Further, recent work [38] has demonstrated a distinction between genes in abundance of cis-acting and trans-acting variants, reinforcing the importance of phase in interpretation of protein-coding genetic variation.”*(page 13, line 38).

Comment 6. *The general definition of ‘haplotype’ in the Introduction (“Such differences result from the unique combination of DNA variants (genotype) in each individual, inherited together from a single parent”) sounds awkward, needs to be phrased more clearly and checked against definitions established in the literature. Then, the (commonly used) genetic definition of haplotypes as the combinations of variants on each of the two chromosomes (of a gene) is effectively not compatible with the analysis of protein haplotypes defined as the translated transcripts of gene haplotypes (a molecular definition of haplotypes independent of existing variants;*

A6. We have reworked the definition to “An individual’s genome is diploid comprising both the maternal and paternal allelic sequences, i.e. each gene has two haplotypes. We define a protein haplotype as the translation of a spliced RNA transcript derived from a gene haplotype.” (page 1, lines 37-39).

Comment 7. *It should furthermore be stated explicitly that the ‘protein haplotypes’ mean ‘different’, or ‘unique’ haplotypes (which include the reference haplotype), and these need to be distinguished from the entirety of haplotypes in a population, the ‘total haplotype count’, equivalent to the total number of genes x 2 x number of genomes/individuals.*

A7. We have clarified in the manuscript when referring to “unique” protein haplotypes/diplotypes to distinguish from total haplotype counts.

Comment 8. *The frequency of occurrence (FoO) of a protein haplotype of a gene needs to be defined, supposedly calculated as: counts of a unique haplotype of a gene in a population, divided by total haplotype count of this gene, equivalent to 2x number of individuals/genomes = 2 x 2504 = 5008.*

A8. We have added the following definition of FoO: “calculated as the count of a unique haplotype of a gene in a population, divided by total haplotype count (twice the number of individuals in the population).” (page 2, lines 21-23).

Comment 9. *Also, it has remained unclear to me, whether a priori only genes with mutations annotated by Ensembl’s Variant Effect Predictor are included in analysis, or all genes with amino acid exchanges, and to which extent/at which stage, genes with functionally protein-altering mutations predicted by PolyPhen-2 and SIFT are included. More information on the choice and use of mutation algorithms is needed.*

A9. We anticipate that the additional description of the Haplosaurus method that we now provide (page 5, lines 3-12) will help in this regard.

Comment 10. *given that a certain fraction of genes are invariable, it is not clear whether the average numbers of protein haplotypes per gene are calculated across all (variable and invariable) genes, or across variable genes only.*

A10. This is an important point; numbers of protein haplotypes per gene are calculated across both variable and invariable genes. To remove the invariable genes from such calculations is misleading; even the invariable genes are most likely variable if you have large enough populations. We now note: *“Our calculation includes all haplotypes, including the reference sequence (where it occurs), and all genes, both variable and invariable.”* (page 5, lines 31-33).

***Comment 11.** The rationale underlying the selection of the most frequent protein haplotypes in the population as the basis for individualized drug design needs to be outlined more coherently at the beginning. If ‘frequency of occurrence’ of a protein haplotype guides the selection, this apparently may work for drugs, which do not target the specific causative, disease-related protein, which presumably is rare in the population, and effectively requires knowledge of the genetic risk profile. This key point has been addressed in examples, bits and pieces all over the manuscript, at best probably in discussion. It should be presented, however, comprehensibly in the Introduction.*

A11. The reviewer raises another important point that we now clarify at the start of the introduction: *“We focus this paper on population-level, genome-wide distributions of common protein haplotypes that can potentially impact drug binding, rather than the specific (and often rare) haplotypes that cause diseases or act as marker proteins.”* (page 1, lines 33-36).

***Comment 12.** In diploid genomes, gene/protein functions are mediated by two forms of the gene/protein, i.e., by pairs of haplotypes (diplotypes). The specific role of each of these two haplotypes remains to be largely clarified, as does their (differential) expression status. Drysdale et al. (PNAS 2000; 97: 10483-10488) have, for example, demonstrated the importance of diplotypes over haplotypes in relation to drug response. These issues and their impact on the authors’ approach have not been addressed at all.*

A 12. We agree that the original manuscript was somewhat dismissive of diplotypes. To the introduction we have added: *“Although the target diplotype rather than haplotype is the ultimate determinant of activity of a drug in an individual only the haplotype need be considered [in drug development] as a drug effective for 99% of protein haplotypes in a population is statistically guaranteed for 98% of diplotypes”.* (page 2, lines 26-27 and page 3, line 1).

However, we do recognise the importance of diplotypes, and now refer to diplotypes in multiple places throughout the revised manuscript (page 1, lines 39-41; page 2 lines 3 and 26; page 6, lines 1-10; page 10, lines 23 and 33; page 12, line 26, page 13, line 33).

Response to Reviewer #3

We thank the Reviewer for their insightful comments, and overall enthusiasm for our approach. We address specific comments below.

***Comment 1.** The examples shown illustrate the potential usefulness, though the added value of analysis by haplotype is not obvious (compared with analysis of individual amino acids).*

A1. This is a good point which we address by adding the following to the discussion section: “information from protein haplotypes and their diplotype pairs enables the functional importance of phase to be assessed, information that is ignored in the analysis of individual protein-coding variants.” (page 12, lines 25-28) and “where alleles of two different variants that each independently inhibit drug activity; in this case haplotype resolution would be required to predict drug activity according to whether an individual with both alleles carried them on the same (cis-acting) or different (trans-acting) chromosomes of a pair.” (page 13, lines 34-37).

***Comment 2.** I do not understand why the authors persistently use the strange acronym FoO, when the perfectly good word "frequency" already exists. Similarly (line 127)*

A2. The definition of “frequency of occurrence” and its acronym “FoO” was introduced by Hoehe et al. (Nat Commun 2014;5:5569). As an important work in the field we decided to follow their terminology. We have made this explicit in the introduction: “...the frequency at which each unique protein haplotype occurs within a population (frequency of occurrence, FoO, following the terminology of Hohe et al)...” (page 2, lines 19-20).

***Comment 3.** the word "significant" should be reserved for statistically significant observations, and its meaning should not be changed to indicate classification by frequency.*

A3. We concur, and have replaced the term “significant haplotype” with “common haplotype” having noted that “we limit many of our analyses to common protein haplotypes that we define as those that equal or exceed a threshold frequency of occurrence (FoO) of 1%” (page 6, lines 5-6).

***Comment 4.** Overall, it will detract from the clarity and reach of the paper to describe as significant or FoO >1% haplotypes which would conventionally be simply described as "MAF >0.01".*

A4. We would argue that minor allele frequency (MAF), referring to the frequency at which the second most common allele occurs in a given population, is materially different from the frequency at which a haplotype occurs in a population (FoO). We therefore argue that no change is needed to the manuscript.

***Comment 5.** The first sentence of the Introduction (lines 27-28) declares that*

there are human protein haplotypes that differ at hundreds of amino acid positions. I find it very hard to believe that this is literally true, and I would be interested to see a single example.

A5. The assertion is *literally* true in, as in ACTN3-001:211R>Q,577R>*_{578del}(325), which is an example provided in the manuscript that differs from the reference at 325 residues following the stop codon. However, we recognise this is atypical and that the statement is somewhat misleading so we have removed it.

Comment 6. *Looking at the genes (Supplementary Table 1) listed as having the largest number of variant haplotypes, many of the highest-scoring genes (like FLG, ANHAK2 and the mucin genes MUC4, 2, 16, 12 etc.) are famous for containing variably repeated coding sequences, and I think it is very likely that the extreme apparent divergences between haplotypes in these genes are artefacts of misalignment rather than truly high frequencies of missense (substitutional) variation.*

A6. This is correct, and well worth distinguishing from the MHC genes where variation is likely to be real and functionally important. To that end we have added the following; *“Mucins are recent paralogues and contain stretches of variably repeated coding sequences which makes sequence alignment and variant calling error prone [<https://doi.org/10.1093/bioinformatics/btx133>, <https://doi.org/10.1038/srep31791>]; it is probable that the extreme apparent divergences between haplotypes in these genes are artefacts of misalignment rather than truly high frequencies of missense (substitutional) variation.”* (page 6, lines 31-35).

Comment 7. *Finally, it is noteworthy that there is almost no description of methodology in the paper itself, with the description of how Haplosaurus was put together relegated to Supplementary Notes. I think that at the very least there should be an outline description of how phased 1000 Genomes Project data were integrated into the study.*

A7. The original description of the methodology was placed in the Supplementary Notes due to word length limitations. We acknowledge that it would be better placed in the main manuscript, and have now accommodated the methodology in the revised main manuscript as follows: *“For a given gene identifier and VCF file, Haplosaurus retrieves the corresponding gene model from a linked Ensembl database. Next, the phased variants that overlap the gene's location are retrieved from the VCF file and used to generate two lists of DNA sequence alleles for each sample, one for each DNA haplotype, according to their phase. The DNA haplotype sequences are reconstructed by substituting the alleles for each haplotype into the reference sequence according to their genomic location. The DNA haplotypes are virtually 'transcribed' to coding sequence (CDS) haplotypes, and these are then virtually 'translated' into protein haplotypes. Once the protein haplotypes for many samples have been generated, the software can calculate the FoO of each unique protein haplotype sequence in a population.”* (page 5, lines 3-12). We then note: *“To enable genome-wide analysis of protein haplotype diversity, we used Haplosaurus to build a database of unique protein haplotypes from phased haplotypes imported directly from the 1000 Genomes Project phase 3 VCF file (Supplementary Note 3).”* (page 5, lines 20-29).

Comment 8. *The Supplementary Note implies that phased haplotypes were directly imported from the 1000 Genomes Project; that is likely to be reliable for most genes, but I found it difficult to imagine how the correct protein haplotypes could be reconstructed from doubly heterozygous individuals for genes that crossed recombination hotspots. Presumably in such cases there will be free association between different LD blocks, but I think this is a non-trivial question that deserves at least some comment in the main paper.*

A8. We have added a note: “Haplotype counts are likely to be reliable for most genes, but overestimated for genes in regions where phasing is difficult (e.g. low linkage disequilibrium) or variant calling is error prone (e.g. repetitive regions)”. (page 5, lines 34-36).

Reviewer #1 (Remarks to the Author):

I have no further comments and feel that the authors addressed my concerns adequately.

Reviewer #2 (Remarks to the Author):

The reviewer acknowledges the authors' efforts to comprehensively rework parts of the manuscript, addressing the reviewer's concerns. The manuscript has been substantially improved in these parts. In order to give final consent to publication, few 'last edits' need to be incorporated to give full credit to the work on 'protein haplotypes' by Hoehe et al., Nat Commun, 2014. This group has introduced this term into the recent human genome literature, adding 'protein haplotypes' to sequence haplotypes', i.e. the translational level.

Page 1, lines 31 - 33: "Whilst there are many genome-wide tools and databases for protein isoforms [3, 4] and genomic variation [5, 6], there is an almost complete absence of resources for protein haplotypes." Modify to "...there is, with rare exceptions (Hoehe et al., Nat Commun, 2014), an almost complete absence of resources for protein haplotypes." This group has made f.i. some gene sets with protein haplotypes (covering the 'Common Diploypic Proteome' they described) available.

Page 1, lines 39 - 40: "In diploid genomes, function is mediated by two forms of the gene/protein, i.e., by pairs of haplotypes (diplotypes)". Add also here the citation Hoehe et al., Nat Commun, 2014. This group has emphasized and discussed this fact again and again in their work. This means switching reference numbers 7 and 8.

Page 1, lines 43-45: "The groundwork for protein haplotype and diplotype architecture of human genomes was laid by Hoehe et al [8] who described a systematic, quantitative, population-based analysis of protein haplotypes in European populations." Modify to "... who described a first systematic, quantitative, population-based analysis of protein haplotypes..."

Reviewer #3 (Remarks to the Author):

I found it very difficult to evaluate this revised version because the pages of the revised manuscript were not numbered, and the line numbers did not correspond to the line numbers used to specify inserted text in the response letter's description. There are even instances in which the text quoted in the response letter does not correspond exactly to the real text in the revised manuscript. For example, the text quoted in the letter as being on page 5, lines 20-29(?) is "To enable genome-wide analysis of protein haplotype diversity, we used Haplosaurus to build a database of unique protein haplotypes from phased haplotypes imported directly from the 1000 Genomes Project phase 3 VCF file (Supplementary Note 3)", whereas the text on page 5 (lines numbered 146-148) actually has "To enable genome-wide analysis of protein haplotype diversity, we used Haplosaurus to build a database of unique protein haplotypes from phased haplotypes imported directly from the 1000 Genomes Project phase 3 dataset (Supplementary Note 3)." The difference is small ("VCF file" instead of "dataset"), but I find it worrying for the process of my rereview that these sentences are not identical.

Having said that, I think that the responses described appear to be proportionate and reasonable, and if (see above) the changes specified in the response letter have really been incorporated at all relevant points in the new text, I think they are satisfactory.

Author's response to Reviewers' comments

Dear Editor,

We have addressed each of the specific comments raised by the three reviewers. Each comment, and how we have incorporated changes into the revised manuscript, are described below

REVIEWERS' COMMENTS:

Response to Reviewer #1 (Remarks to the Author):

We thank the Reviewer for their comments and acceptance of the revisions.

Comment 1: *"I have no further comments and feel that the authors addressed my concerns adequately."*

A1. We thank the reviewer for their acknowledgment of our work to incorporate their previous comments.

Response to Reviewer #2 (Remarks to the Author):

We appreciate and thank the Reviewer for their thoughtful comments.

Comment 1: *"The reviewer acknowledges the authors' efforts to comprehensively rework parts of the manuscript, addressing the reviewer's concerns. The manuscript has been substantially improved in these parts. In order to give final consent to publication, few 'last edits' need to be incorporated to give full credit to the work on 'protein haplotypes' by Hoehe et al., Nat Commun, 2014. This group has introduced this term into the recent human genome literature, adding 'protein haplotypes' to sequence haplotypes', i.e. the translational level."*

A1. We thank the reviewer for their insightful comments and appreciation of our work. We agree with the Reviewer's comments and have revised the manuscript incorporating the Reviewer's suggestions. Specific comments are addressed below.

Comment 2: *"Page 1, lines 31 - 33: "Whilst there are many genome-wide tools and databases for protein isoforms [3, 4] and genomic variation [5, 6], there is an almost complete absence of resources for protein haplotypes." Modify to "...there is, with rare exceptions (Hoehe et al., Nat Commun, 2014), an almost complete absence of resources for protein haplotypes." This group has made f.i. some gene sets with protein haplotypes (covering the 'Common Diplotypic Proteome' they described) available."*

A2. Edited as suggested on Page2, lines 1-3: *"Whilst there are many genome-wide tools and databases for protein isoforms^{3,4} and genomic variation^{5,6}, there is, with rare exceptions⁷, an almost complete absence of resources for protein haplotypes."*

Comment 3: *"Page 1, lines 39 - 40: "In diploid genomes, function is mediated by two forms of the gene/protein, i.e., by pairs of haplotypes (diplotypes)". Add also here the citation Hoehe et al., Nat Commun, 2014. This group has emphasized and discussed this fact again and again in their work. This means switching reference numbers 7 and 8."*

A3. Edited as suggested on Page2, lines 9-11 : “In diploid genomes, function is mediated by two forms of the gene/protein, i.e., by pairs of haplotypes (diplotypes)⁷.”

Comment 4: “Page 1, lines 43-45: “The groundwork for protein haplotype and diplotype architecture of human genomes was laid by Hoehe et al [8] who described a systematic, quantitative, population-based analysis of protein haplotypes in European populations.” Modify to “... who described a first systematic, quantitative, population-based analysis of protein haplotypes...”

A4. Edited as suggested on Page2, lines 13-17 : “The groundwork for protein haplotype and diplotype architecture of human genomes was laid by Hoehe et al⁷ who described a first systematic, quantitative, population-based analysis of protein haplotypes.”

Response to Reviewer #3 (Remarks to the Author):

We thank the Reviewer for their detailed comments and approval of the revised manuscript. We apologise for the difficulty in the reviewing process by omitting line numbers in the manuscript. This has been rectified and will be adopted in the future. The previously raised comments in our answers to reviewer’s letter can be accessed as follow with corrected Page and Lines numbering due to text final revision.

Previous Comment 1. The examples shown illustrate the potential usefulness, though the added value of analysis by haplotype is not obvious (compared with analysis of individual amino acids).

Edited A1. This is a good point which we address by adding the following to the discussion section: “information from protein haplotypes and their diplotype pairs enables the functional importance of phase to be assessed, information that is ignored in the analysis of individual protein-coding variants.” (page 13, lines 3-5) and “where alleles of two different variants that each independently inhibit drug activity; in this case haplotype resolution would be required to predict drug activity according to whether an individual with both alleles carried them on the same (cis-acting) or different (trans-acting) chromosomes of a pair.” (page 13, lines 41-42).

Previous Comment 2. I do not understand why the authors persistently use the strange acronym FoO, when the perfectly good word "frequency" already exists. Similarly (line 127)

Edited A2. The definition of “frequency of occurrence” and it’s acronym “FoO” was introduced by Hoehe et al. (Nat Commun 2014;5:5569). As an important work in the field we decided to follow their terminology. We have made this explicit in the introduction: “...the frequency at which each unique protein haplotype occurs within a population (frequency of occurrence, FoO, following the terminology of Hohe et al)...” (page 2, lines 35-36).

Previous Comment 3. the word "significant" should be reserved for statistically significant observations, and its meaning should not be changed to indicate classification by frequency.

Edited A3. We concur, and have replaced the term “significant haplotype” with “common haplotype” having noted that “we limit many of our analyses to common protein haplotypes that we define as those that equal or exceed a threshold frequency of occurrence (FoO) of 1%” (page 6, lines 5-6).

Previous Comment 4. Overall, it will detract from the clarity and reach of the paper to describe as significant or FoO >1% haplotypes which would conventionally be simply described as "MAF >0.01".

Edited A4. We would argue that minor allele frequency (MAF), referring to the frequency at which the second most common allele occurs in a given population, is materially different from the frequency at which a haplotype occurs in a population (FoO). We therefore argue that no change is needed to the manuscript.

Previous Comment 5. The first sentence of the Introduction (lines 27-28) declares that there are human protein haplotypes that differ at hundreds of amino acid positions. I find it very hard to believe that this is literally true, and I would be interested to see a single example.

Edited A5. The assertion is *literally* true in, as in ACTN3-001:211R>Q,577R>*,578del(325), which is an example provided in the manuscript that differs from the reference at 325 residues following the stop codon. However, we recognise this is atypical and that the statement is somewhat misleading so we have removed it.

Previous Comment 6. Looking at the genes (Supplementary Table 1) listed as having the largest number of variant haplotypes, many of the highest-scoring genes (like FLG, ANHAK2 and the mucin genes MUC4, 2, 16, 12 etc.) are famous for containing variably repeated coding sequences, and I think it is very likely that the extreme apparent divergences between haplotypes in these genes are artefacts of misalignment rather than truly high frequencies of missense (substitutional) variation.

Edited A6. This is correct, and well worth distinguishing from the MHC genes where variation is likely to be real and functionally important. To that end we have added the following; “Mucins are recent paralogues and contain stretches of variably repeated coding sequences which makes sequence alignment and variant calling error prone [<https://doi.org/10.1093/bioinformatics/btx133>, <https://doi.org/10.1038/srep31791>]; it is probable that the extreme apparent divergences between haplotypes in these genes are artefacts of misalignment rather than truly high frequencies of missense (substitutional) variation.” (page 6, lines 30-34).

Previous Comment 7. Finally, it is noteworthy that there is almost no description of methodology in the paper itself, with the description of how Haplosaurus was put together relegated to Supplementary Notes. I think that at the very least there should be an outline description of how phased 1000 Genomes Project data were integrated into the study.

Edited A7. The original description of the methodology was placed in the Supplementary Notes due to word length limitations. We acknowledge that it would be better placed in the main manuscript, and have now accommodated the methodology in the revised main manuscript as follows: *“For a given gene identifier and VCF file, Haplosaurus retrieves the corresponding gene model from a linked Ensembl database. Next, the phased variants that overlap the gene’s location are retrieved from the VCF file and used to generate two lists of DNA sequence alleles for each sample, one for each DNA haplotype, according to their phase. The DNA haplotype sequences are reconstructed by substituting the alleles for each haplotype into the reference sequence according to their genomic location. The DNA haplotypes are virtually ‘transcribed’ to coding sequence (CDS) haplotypes, and these are then virtually ‘translated’ into protein haplotypes. Once the protein haplotypes for many samples have been generated, the software can calculate the FoO of each unique protein haplotype sequence in a population.”* (page 5, lines 5-14). We then note: *“To enable genome-wide analysis of protein haplotype diversity, we used Haplosaurus to build a database of unique protein haplotypes from phased haplotypes imported directly from the 1000 Genomes Project phase 3 VCF file (Supplementary Note 3).”* (page 5, lines 22-24).

New Comment 1: *“I found it very difficult to evaluate this revised version because the pages of the revised manuscript were not numbered, and the line numbers did not correspond to the line numbers used to specify inserted text in the response letter’s description. There are even instances in which the text quoted in the response letter does not correspond exactly to the real text in the revised manuscript. For example, the text quoted in the letter as being on page 5, lines 20-29(?) is “To enable genome-wide analysis of protein haplotype diversity, we used Haplosaurus to build a database of unique protein haplotypes from phased haplotypes imported directly from the 1000 Genomes Project phase 3 VCF file (Supplementary Note 3)”, whereas the text on page 5 (lines numbered 146-148) actually has “To enable genome-wide analysis of protein haplotype diversity, we used Haplosaurus to build a database of unique protein haplotypes from phased haplotypes imported directly from the 1000 Genomes Project phase 3 dataset (Supplementary Note 3).” The difference is small (“VCF file” instead of “dataset”), but I find it worrying for the process of my rereview that these sentences are not identical.*

Having said that, I think that the responses described appear to be proportionate and reasonable, and if (see above) the changes specified in the response letter have really been incorporated at all relevant points in the new text, I think they are satisfactory.”

A1. We agree with the reviewer’s comment on the change of text from “VCF file” instead of “dataset” which has been an editing error on our part. It has been edited accordingly on Page 5, lines 22-24: *“To enable genome-wide analysis of protein haplotype diversity, we used Haplosaurus to build a database of unique protein haplotypes from phased haplotypes imported directly from the 1000 Genomes Project phase 3 VCF file (as described in Methods).”*